

# Halogen chemistry in volcanic plumes: a 1D framework based on MOCAGE-1D (version R1.18.1) preparing 3D global chemistry modelling

Virginie Marécal[1], Ronan Voisin-Plessis[1], Tjarda J. Roberts[2], Alessandro Aiuppa[3], Herizo Narivelo[1], Paul D. Hamer[4], Béatrice Josse[1], Jonathan Guth[1], Luke Surl[2,5,6], and Lisa Grellier[1,*]

[1]Centre National de Recherches Météorologiques, Université de Toulouse, Météo-France, CNRS, Toulouse, 31000, France
[2]Laboratoire de Physique et Chimie de l'Environnement et de l'Espace, UMR7328, CNRS-Université d'Orléans, 45000, France
[3]Dipartimento DiSTeM, Università di Palermo, Palermo, 90123, Italy
[4]NILU – Norwegian Institute for Air Research, P.O. Box 100, Kjeller, 2027, Norway
[5]LATMOS/IPSL, Sorbonne Université, UVSQ, CNRS, Paris, France
[6]Department of Chemistry, University of Aberdeen, Aberdeen, UK
[*]now at Citepa, Paris, France

*Correspondence to*: Virginie Marécal (virginie.marecal@meteo.fr)

**Abstract.** Volcanoes are a known source of halogens to the atmosphere. HBr volcanic emissions lead rapidly to the formation of BrO within volcanic plumes. BrO, having a longer residence time in the atmosphere than HBr, is expected to have an impact on tropospheric chemistry, at least at the local and regional scales. The objective of this paper is to prepare a framework for further 3-D modelling of volcanic halogen emissions in order to determine their fate within the volcanic plume and then in the atmosphere at the regional and global scales. This work is based on a 1-D configuration of the global chemistry transport model MOCAGE whose low computational cost allows us to perform a large set of sensitivity simulations. This paper studies the Mount Etna eruption on 10 May, 2008. Several reactions are added to MOCAGE to represent the halogen chemistry occurring within the volcanic plume. A simple sub-grid scale parameterization of the volcanic plume is also implemented and tested. The use of this parameterization tends to limit slightly the efficiency of BrO net production. Both simulations with and without the parameterization give similar results for the partitioning of the bromine species, ozone depletion and of the BrO/SO₂ ratio that are consistent with previous studies and with the BrO/SO₂ ratio in the volcanic plume estimated from GOME-2 spaceborne observations.

A series of test experiments were performed to evaluate the sensitivity of the results to the composition of the emissions, and, in particular, primary sulphate aerosols, the Br radical, and NO. Simulations show that the plume chemistry is sensitive to these assumptions. Another series of tests on the effective radius assumed for the volcanic sulphate aerosols shows that BrO net production is sensitive to this parameter with lower BrO concentrations reached when larger aerosols (smaller total surface





area) are assumed. We also find that the maximum altitude of the eruption changes the BrO production, which is linked to the vertical variability of the concentrations of oxidants. These sensitivity tests display changes in the bromine chemistry cycles that are generally at least as important as the subgrid scale plume parameterization.

Overall, the version of the MOCAGE chemistry developed for this study is suitable to produce the expected halogen chemistry in volcanic plumes during daytime and night. These results will be used to guide the implementation of volcanic halogen emissions in the 3-D configuration of MOCAGE for regional and global simulations.

# 1 Introduction

Volcanoes are an important source of gases injected into the atmosphere. In addition to the main gaseous emissions of water
vapour, $CO_2$, and $SO_2$, volcanoes also emit inorganic halogen compounds mainly as HCl, HF and HBr (Gerlach, 2004). HF is very unreactive in the context of gas phase tropospheric chemistry, while HCl and HBr are both reactive species in this environment. Bromine, and to a lesser extent chlorine, induce tropospheric ozone loss at the global scale and subsequent OH loss, therefore affecting the tropospheric oxidising capacity (e.g. Saiz-Lopez and von Glasow 2012, Simpson et al. 2015, Sherwen et al., 2016). But the hydrogen halides (HX, with X=Cl, Br, F, I) have a high effective solubility meaning that HCl
and HBr emitted by volcanoes are scavenged onto the Earth's surface by wet deposition within a few hours to a few days. Consequently, their direct impact on the air composition in the troposphere was expected to be local and weak.

However, this point of view was challenged when Bobrowski et al. (2003) observed bromine monoxide (BrO) in the plume of the Soufrière Hills volcano, Montserrat. After this first observation, BrO has been measured in many other volcanic plumes (e.g. Oppenheimer et al., 2006; Theys et al., 2009; Boichu et al., 2011; Bobrowski and Giuffrida, 2012; Hörmann et al., 2013,
Kern and Lyons, 2018; Roberts, 2018; Seo et al., 2019). The detection of volcanic BrO is significant because unlike HCl and HBr, BrO is not water-soluble. Its observed presence several kilometres downwind also indicates occurrence of reactive halogen cycling in volcanic plumes from HBr. This implies a longer atmospheric residence time for volcanic bromine, and therefore opens conditions for regional to global scale impacts on tropospheric chemical composition. The purpose of this study is to prepare a framework for simulating the atmospheric chemistry of volcanic halogen emissions in a global model, in
order to determine their fate in the volcanic plume, and ultimately at the regional and global scales.

Regarding the source of volcanic BrO, Gerlach (2004) first suggested that BrO is not directly emitted by volcanoes, and that chemical reactions in the high-temperature mixture of air and magmatic gases, immediately following emission, generate radicals that could potentially form BrO further downwind. A variety of such mixtures, depending on varying proportions of air and magmatic gases, were later studied by Martin et al. (2006). However, this near-vent source of radicals (Br, Cl, NO,
OH) cannot itself explain the occurrence of BrO further downwind. Studies of the multi-phase plume atmospheric chemistry (e.g. Oppenheimer et al., 2006; Bobrowski et al., 2007; Roberts et al., 2009, 2014; von Glasow et al., 2009; Kelly et al., 2013; Surl et al., 2015; Jourdain et al., 2016, Surl et al., 2021) have highlighted autocatalytic reaction cycles as the key mechanism for BrO production in later stages of volcanic plume evolution, at temperatures closer to that of ambient air. Rapid bromine





cycling can also lead to the formation of reactive chlorine (e.g. Jourdain et al., 2016, Roberts et al., 2018). The basis for BrO
formation is halogen heterogeneous chemistry occurring on sulphate aerosol. This process is similar to the so-called ``bromine
explosion'' (Platt and Lehrer, 1997; Wennberg, 1999) that was identified in the tropospheric Polar region. The net reaction of
the cycle consists of a rapid and strong production of BrO. Ozone molecules are depleted during this cycle. The environment
where the chemical cycle takes place needs to have a pH < 7 (Fickert et al., 1999). This pH condition is readily achieved in a
volcanic plume containing acid gases and sulphate aerosols. Moreover, the 'at source' or primary sulphate aerosols present in
the volcanic source promote heterogeneous chemistry to form BrO. Model sensitivity tests (e.g. Roberts et al. 2014) find that
high-temperature radicals (Br, Cl, NO, OH) in the model initialisation act to 'kick-start' the onset of the bromine explosion.
Numerical atmospheric models (e.g. Bobrowski et al. 2007, Roberts et al. 2009) containing the bromine explosion mechanism
and initialised with a volcanic emission that includes HBr, HCl, SO2, primary sulphate, and a representation of the high-
temperature radicals (e.g. Br, Cl, NO, OH) were able to reproduce the BrO observed downwind from volcanoes. More details
on the current knowledge on bromine in volcanic plumes is given in the review by Gutmann et al. (2018).

Most previous numerical modelling studies describing halogen chemistry in volcanic plumes (Bobrowski et al., 2007; Roberts
et al., 2009, 2014; von Glasow, 2010; Kelly et al., 2013; Surl et al., 2015) focused on the local volcanic chemistry within the
plume in a zero or one-dimensional Lagrangian framework. The same thermodynamic equilibrium model was used in the
initialisation of the atmospheric chemistry models in most of these studies (Bobrowski et al., 2007; Roberts et al., 2009; von
Glasow, 2010; Surl et al., 2015) to describe the high temperature mixtures of air and volcanic gases. This model is the ``HSC
Chemistry'' software (Martin et al., 2006; Martin et al., 2009) that, similar to the abovementioned work of Gerlach (2004),
predicts the high-temperature formation of many other species than the raw volcanic emissions, in particular halogen radicals
and oxidants. The plume/atmospheric chemistry modelling studies initialised using HSC outputs show a rapid increase in BrO
within the plume in the few minutes after an emission, consistent with plume observations. However, recent studies (e.g.
Aiuppa et al., 2007a; Martin et al., 2012; Roberts et al., 2019) have shown that the assumption of thermodynamic equilibrium
used in HSC is not realistic, in particular for NOx and $H_2S$. New kinetics-based models of the hot plume chemistry are in
development (Roberts et al., 2019) but do not yet contain halogens.

Most previous studies on the volcanic plume chemistry were at the plume scale over only a few hours from emission. However,
because BrO is not soluble, it can be transported over at least regional scales. It is therefore also interesting to study its effect
at larger time and spatial scales. For this, it is possible to use 3D regional or global atmospheric chemistry models. Jourdain et
al. (2016) studied an episode of extreme passive degassing of Ambrym (Vanuatu) with the coupled meteorology-chemistry
mesoscale model C-CCATT-BRAMS (Longo et al., 2013) with 4 nested grids from 50 km (regional grid) down to 0.5 km
(close to vent grid) horizontal resolutions. Their results confirmed the influence of volcanic halogen emissions at the local and
regional scales on the oxidising capacity of the troposphere. In particular, they showed an impact on methane lifetime.
Recently, Surl et al. (2021) studied a plume from Mt Etna passive degassing based on 3D model simulations with WRF-Chem
model at ~1 km resolution compared to aircraft observations of ozone and ground-based remote sensing of BrO. The study
focused on the region from the volcano to tens of km downwind. Surl et al. (2021) show that the wind speed and the time of





the day have non-linear effects on the BrO/SO$_2$ ratio that characterizes the BrO production efficiency. They also highlight the impacts of the halogen chemistry on reactive nitrogen, and on HOx with consequence of slower secondary sulfate aerosol formation. From sensitivity simulations, they confirmed the importance of the composition of the emission source resulting from high temperature processes, in particular Br radicals, for the rapid BrO production in the plume. Both of these 3D model studies used nested grids to simulate plume chemistry in a regional model at high spatial resolution (km) over a limited area. A step further is to study this influence from the regional to the global scales based on 3D chemistry-transport models (CTMs). Because of the typical coarse resolution of such models (typically from ~2° to ~0.1°, or 100's to 10's km), there is no possibility to represent the fine scale plume chemistry in global CTMs. Processes occurring at sub-grid scales are generally represented *via* parameterizations in atmospheric models, giving a better description of the phenomenon studied in case of plumes (e.g. Karamchandani et al., 2002; Cariolle et al., 2009). Therefore, a parameterization might be required to properly represent the rapid chemistry processing within the volcanic plumes in their early stages when they contain high concentrations of sulfur and halogens. This was one of the aims of the study of Grellier et al. (2014) that developed and tested in a one-dimensional (vertical column) modelling framework a simple subgrid-scale parameterization of halogen plume chemistry at 0.5° and 2° horizontal resolutions. This study was not successful because of its simplified representation of bromine chemistry, in particular the lack of explicit representation of Br$_2$ species. The other aim of the study by Grellier et al. (2014) was to make sensitivity simulations to several input parameters that can affect the bromine explosion.

The present paper is a revised version of Grellier at al. (2014), with the same general aim of preparing from 1D simulations the implementation and use of volcanic halogen chemistry in the 3-D global/regional CTM MOCAGE. The framework is as in Grellier et al. (2014) but extended and with major updates, in particular the introduction of Br$_2$ species and an expanded and more realistic version of the subgrid scale parameterization. Because of the low computing cost of the 1D simulations, we performed a set of sensitivity tests on the impact of different parameters on the bromine cycle within the plume. This includes the choice of the composition of the volcanic emissions used as input. As discussed above, there is not a full understanding of the processes occurring when magmatic air first mixes with atmospheric air at high temperature. Previous studies showed that the choice of this composition is important at fine scale resolutions (Roberts et al., 2009; Roberts et al. 2014, Jourdain et al., 2016; Surl et al., 2021). Here, we will investigate this issue at a coarser horizontal resolution that is typical of the 3D MOCAGE simulations. We also use the 1D MOCAGE modelling framework as a testbed to analyse the impact of the time of the day, the size of the volcanic sulphate aerosols and the altitude of the emissions on the bromine explosion efficiency.

In Sect. 2, a description of the volcanic eruption studied in this paper is given. Then the numerical model, 1D version of MOCAGE, is presented in Sect. 3, including the upgrades needed to represent volcanic halogen chemistry. The simulations are described in Sect. 4. Section 5 presents the analysis of the results of the simulations. Conclusions are given in Sect. 6.





## 2 Case study: the Etna eruption of 10 May 2008

### 2.1 General description

Mount Etna is the most active volcano in Europe and among the largest point sources of volcanic volatiles on the planet (Aiuppa et al., 2008). Gases and aerosols and possibly volcanic ash are continually emitted by the craters by passive or explosive degassing. Four craters are currently hosted on the volcano summit; the volcano itself has a total surface area of 1200 square kilometres and the mean altitude of the volcanic plateau is at an altitude of 3300m.

This study focuses on the eruption of Mount Etna that occurred on 10 May 2008 (see Bonaccorso et al. (2011) for more
information about this eruptive event). There are three reasons behind the choice of this volcano. The first reason is the fact that Mount Etna is one of the largest known emission sources of halogens (Aiuppa et al., 2005). Etna volcano is also continuously and extensively monitored by INGV (Istituto Nazionale di Geofisica e Vulcanologia), and therefore a variety of gas composition information on emissions is available. These data are used to simulate the eruption with the numerical model. The third reason is that satellite observations of the plume have been made above the Mediterranean region. In the supplement
of Hörmann et al. (2013), the tropospheric slant columns of BrO and $SO_2$ in the plume have been retrieved from the GOME-2 instrument on the morning after the eruption.

The eruption on 10 May 2008 that we study started at 14:15 UTC and lasted until 18:15 UTC (from monitoring reports of INGV-Osservatorio Etneo; available at www.ct.ingv.it). The eruptive cloud was injected from the top of Mount Etna 3300m up to about 8500m in altitude above mean sea level (Bonaccorso et al., 2011). The time-averaged $SO_2$ daily release on the day
of the eruption was estimated to be 10,000 tons, which is obtained by averaging results of car traverses made with an Ocean Option USB2000 + spectrometer and DOAS retrieval technique. During May 2008, passive emissions from the volcano contributed an average of 2,000 tons of $SO_2$ per day (from monitoring reports of INGV-Osservatorio Etneo; available at www.ct.ingv.it and from G. Salerno, personal communication, 2013).

### 2.2 Gas composition of the volcanic emissions

The composition of the gas plume released by Mount Etna has extensively been characterised in years before the selected case study by both in situ (e.g., Aiuppa et al., 2007b, 2008) and remote sensing (Allard et al., 2005) techniques. These studies have shown that, as for volcanic gas emissions in general (Oppenheimer, 2003), Etna's magmatic volatiles are dominated by $H_2O$, $CO_2$ and $SO_2$, in proportions varying both in time (depending on activity state) and space (e.g. from crater to crater). Etna's magmatic gases also include smaller but significant amounts of halogen species (HCl, HF and HBr).

Remote sensing techniques (e.g., Fourier-Transform Infrared Spectroscopy, FTIR), that can be operated from more distal -- safer -- locations, are inherently more appropriate to investigate the compositional gas features during eruptions. Passive open-path FTIR, in particular, is often used to study the chemistry of gas jets propelled by lava fountains at Etna (Allard et al., 2005). Unfortunately, no similar data is available for the 10 May 2008 eruptive episode. Bromine, which is emitted by magmatic systems in the HBr form (Gerlach, 2004), is systematically below the detection threshold of FTIR. Few reports of near-





downwind volcanic BrO (a product of HBr oxidation; Oppenheimer et al., 2006; Bobrowski et al., 2007) during the Etna paroxysms are available, but none for 10 May 2008. Passive (non-eruptive) Br emissions from volcanoes can however be satisfactorily derived by direct sampling of both fumaroles (Gerlach, 2004) and plumes (Aiuppa et al., 2005). This is why we use the magmatic gas composition for the Etna's passive plume (Table 1), derived on 14 May 2008 by a combination of techniques (MultiGAS for $H_2O$, $CO_2$ and $SO_2$ and filter packs for halogens; see Aiuppa et al., 2005, 2007b, 2008 for analytical

details), as an analogue for 10 May 2008 eruptive plume composition. This assumption is motivated by the hydraulic continuity between the central craters (where passive emissions concentrate) and the Southeast crater (the eruptive vent), for which there is plenty of seismic (Patanè et al., 2003), gas (Aiuppa et al., 2010) and infrasonic (Marchetti et al., 2009) evidence. Moreover, since the aim of the paper is to use this case study as a testbed for plume chemistry modelling and not to make a detailed analysis of the eruption, the gas composition on 14 May 2008 being representative of Etna emissions is realistic enough to be

used here.

**Table 1 : Molar ratio percentage of the main species emitted by Mount Etna volcano (magmatic gas composition) on 14 May 2008.**

| | |
|---|---|
| $H_2O$ | 90 |
| $CO_2$ | 8 |
| $SO_2$ | 0.7 |
| HCl | 0.21 |
| HF | 0.09 |
| HBr | $2.3\ 10^{-4}$ |
| HI | $5.4\ 10^{-6}$ |
| $H_2S$ | $4.45\ 10^{-3}$ |
| $H_2$ | 0.162 |
| CO | $2.18\ 10^{-3}$ |

## 3 Model description

The numerical model used for the simulations is a 1-D configuration, called hereafter MOCAGE-1D, of the three-dimensional

global and regional chemistry transport model MOCAGE (Model Of atmospheric Chemistry At larGE scale, Josse et al., 2004, Cussac et al. 2020, Lamotte et al., 2021) that is developed by Météo-France to simulate air composition for research (e.g. Lacressonnière et al., 2014) and operational applications (e.g. Marécal et al., 2015). Due to the low computational cost, this one-dimensional configuration allows us to make a large set of sensitivity tests on the many parameters that can modify the chemical processing within a volcanic plume. It does not intend to reproduce the exact chemical evolution focusing on local

scale at the very early stage (< 1 hour) within the volcanic plume as done in previous studies (e.g. Roberts et al. 2009) but to prepare the configuration for 3-D CTM simulations at the regional or global scales.



The 1-D configuration corresponds here to the vertical column above the emission source (i.e., above Etna's location). Like in the 3-D version of MOCAGE (called hereafter MOCAGE-3D), the vertical resolution is divided into 47 levels from the ground up to 5 hPa. It uses the sigma hybrid coordinate: close to the surface, the levels follow the orography while the highest levels follow isobars. The interval between levels increases with altitude with 7 levels within the planetary boundary layer, 20 in the free troposphere and 20 in the stratosphere.

The 1D configuration also assumes no transport horizontally and vertically (unlike the 3D version). Thus, the boxes constituting the vertical column are not interacting with each other and can be considered as an ensemble of independently piled 0D boxes. Because there is no horizontal transport in MOCAGE-1D, there is no dispersion of the quantities emitted outside the considered column and no mixing with the background air surrounding this column. Even if in reality there is mixing of the volcanic plumes with background air at a scale larger than the model gridbox (here 0.5° longitude x 0.5° latitude), this becomes significant only after several hours up to 1-2 days. Since the MOCAGE-1D simulations are run over a maximum of 20 hours, this setup is thus reasonable to study the plume chemistry and to address its sensitivity to different parameters. The possible impact of neglecting this effect is taken into account in the analysis of the results.

The 1-D configuration of MOCAGE is designed with the future aim to implement the chemical evolution of bromine compounds from volcanic eruptions in MOCAGE-3D. Thus, the injection of the emitted gases during the eruption is done as in MOCAGE-3D by adding volcanic gas amounts to the background air in the gridbox containing the volcano vent and at the model levels impacted by the volcanic plume. For eruptions, an "umbrella" profile is used from the volcano crater altitude to the top height of the plume as in Lamotte et al. (2021) that represents the injection of 75% of the emissions in the top third of the plume.

The chemical reactions represented in MOCAGE-1D are the same as for MOCAGE-3D, i.e. including both the tropospheric and stratospheric chemistry, but with the addition of several reactions necessary to model the bromine explosion in volcanic plumes. The standard version of the model uses the "RACMOBUS" chemistry scheme which is a merge of the REPROBUS stratospheric scheme (Lefèvre et al., 1994) with RACM tropospheric chemistry scheme (Stockwell et al., 1997), completed with the sulfur cycle (Ménégoz et al. 2009, Guth et al. 2016) and peroxyacetylnitrate (PAN) photolysis. RACMOBUSis valid for remote to polluted conditions and from the Earth's surface up to the stratosphere. Altogether, the original version of the model contains 112 species with 316 gaseous reactions and 54 photolysis applied in both the troposphere and the stratosphere, and 9 heterogeneous reactions only applied in the stratosphere. The chemical solver is based on a semi-implicit Euler-backward method.

This scheme has been extended to represent the "bromine explosion" cycle. This cycle is described in detail for instance in Oppenheimer et al. (2006), Platt and Hönninger (2003), Bobrowski et al. (2007) or Roberts et al. (2009) and its main reactions are listed below:

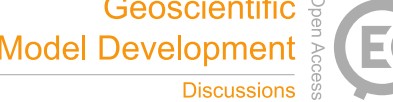

$$HBr + OH \rightarrow Br + H_2O \tag{R1}$$

$$Br + O_3 \rightarrow BrO + O_2 \tag{R2}$$

$$BrO + HO_2 \rightarrow HOBr + O_2 \tag{R3}$$

$$BrO + NO_2 \rightarrow BrONO_2 \tag{R4}$$

$$HOBr + HBr(sulphate\ aerosol) \rightarrow Br_2 + H_2O \tag{R5a}$$

$$HOBr + HCl(sulphate\ aerosol) \rightarrow BrCl + H_2O \tag{R5b}$$

$$BrONO_2 + H_2O(sulphate\ aerosol) \rightarrow HOBr + HNO_3 \tag{R6}$$

$$Br_2 + h\nu \rightarrow 2Br \tag{R7}$$

$$BrCl + h\nu \rightarrow Br + Cl \tag{R8}$$

$$Br + HO_2 \rightarrow HBr + O_2 \tag{R9}$$

$$BrO + BrO \rightarrow 2Br + O_2 \tag{R10}$$

The notation "$HBr(sulphate\ aerosol)$" (resp. "$HCl(sulphate\ aerosol)$") means that HOBr reacts heterogeneously with HBr (resp. HCl) in sulphate aerosols. (R6) corresponds to BrONO$_2$ hydrolysis.

This cycle leads to the autocatalytic BrO formation summarised below:

$$BrO + HO_2 + HBr(sulphate\ aerosol) + 2O_3 \rightarrow 2BrO + 3O_2 + H_2O \tag{R11}$$

$$BrO + NO_2 + HBr(sulphate\ aerosol) + 2O_3 \rightarrow 2BrO + 3O_2 + HNO_3 \tag{R12}$$

Volcanic emissions contain halogen species and in particular HBr that provides the bromine atoms (R1) needed for the cycle to produce BrO. To simulate the bromine explosion cycle, we have modified the halogen chemistry scheme in MOCAGE, originally designed for stratospheric chemistry only. In Grellier et al. (2014), Br$_2$ was assumed to be converted into Br

instantaneously. This assumption being only possibly valid during daytime because of Br$_2$ photolysis, the results of Grellier et al. (2014) simulations were not realistic at night-time. Here we introduced Br$_2$ as a new species and its photolysis (R7) and gas-phase reaction with OH. Additionally, we included the 3 heterogeneous reactions (R5a), (R5b) and (R6) and 6 halogen gaseous reactions following Surl et al. (2021). The Supplement gives the list of the halogen species and reactions present in the updated version of MOCAGE chemistry and details on the calculation of the heterogeneous reactions.

## 4 Setup of the simulations


A large set of 1D simulations was run in different conditions using the Etna case study as a testbed to assess the model ability to produce BrO from HBr volcanic emissions, the impact of using an expanded version of the subgrid scale parameterization of Grellier et al. (2014) and the sensitivity of the bromine explosion to several parameters.

### 4.1 General model setup and description of the reference simulations


Each 1D simulation calculates the chemical concentrations of all species in the vertical levels of the model. The horizontal box size chosen is 0.5° (longitude) x 0.5° (latitude) resolution (~44 x ~55 km$^2$ at the location of Mount Etna), because it is an





intermediate horizontal resolution used both for global and regional studies with MOCAGE. The initial conditions of the chemical species of all simulations are the same. They correspond to the 1-D profile of the species concentrations on 10 May 2008 at 14:00 UTC, extracted from the grid box that contains Mount Etna, in a 3-D MOCAGE global simulation at 0.5°x0.5°

resolution.  In the 1D simulation we set to zero the concentrations for the inorganic chlorine and bromine species in the troposphere. This is done because in the standard version of MOCAGE 3D that is used for the initial conditions, the halogen inorganic species are only used to represent stratospheric chemistry, and their background concentrations in the troposphere cannot be considered to be reliable. Furthermore, the inorganic halogen concentrations are dominated by the injection of the volcanic eruption on the scale of the study. Also, since the focus is on the chemical processing of the eruption emissions in the

plume, we choose to initialise the concentrations of sulphate to zero to quantify only the impact of volcanic sulphate concentrations in our analysis.

The 1-D simulations are run from 10 May 2008 at 14:00 UTC to 11 May 2008 at 10:00 UTC. This includes the time of the satellite observations of BrO and $SO_2$ in the plume gathered in the morning of the day after the eruption (~8:40 UTC) by GOME-2 (Hörmann et al., 2013). The results of our simulations are compared to these data. The meteorological parameters

used in all the 1D simulations are the same and come from ERA-Interim meteorological reanalyses. The same reanalyses are used for the MOCAGE-3D simulation used for the initialisation of the chemical concentrations.

The gaseous composition of the emissions leaving the vent is given in Table 1. We assume a total eruptive $SO_2$ output of 8000 tonnes (cumulative output of 4 h, see section 2.2). The emissions are set from 3300m (crater height) to 8500m above sea level, the top of the plume being estimated from Bonaccorso et al. (2011). The time step for the injection of the emissions is 15 min

as in the MOCAGE-3D version.

Regarding the composition of the volcanic emissions, we need to account for the modification of the volcanic emissions when magmatic gases first mix with ambient air at high temperature. The processes occurring at high temperature are not yet fully known and quantified as discussed in previous sections. Previous modelling studies showed that emissions of primary (or 'at source') volcanic sulfate aerosols and radicals such as Br, Cl or OH are necessary to provide the kick-start to the bromine

explosion in the early stage of the plume (e.g., Roberts et al., 2009; Surl et al. 2021). The primary sulfate aerosols provide surface area to catalyse the bromine heterogeneous chemistry. The Br radicals provide an initial reactive halogen source that 'kick-starts' the halogen chemistry. Br radicals may be produced both directly from high temperature processes and indirectly from reactions involving HBr and high temperature-produced HOx. As in previous studies (e.g. Roberts et al. 2009, Jourdain et al. 2016, Surl et al. 2021) we take into account these changes from high temperature processes in the volcanic emissions

used in MOCAGE 1D-simulations. The molar ratios and associated mass fluxes for the eruption introduced as input to the 1D MOCAGE model are given in Table 2. For $H_2O$, HCl, $H_2S$ and CO, their molar ratio to $SO_2$ is calculated from Table 1. For the emissions of primary sulphate aerosols, we use the ratios of $SO_2$ proposed by Surl et al. (2021) for their '*main*' model experiment simulating a case of $M^t$ Etna passive degassing in 2012. For bromine, we use the HBr/$SO_2$ ratio from Table 1 to get the total number of bromine moles that are then split into 75% HBr and 25% Br as in Surl et al. (2021). Note that because

$CO_2$, HF and HI are not relevant for the bromine chemistry in volcanic plumes, they are not taken into account in this study.



$H_2$ emission is not introduced since they are negligible with respect to background concentrations. $H_2O$ emissions are not included because water vapour is considered as a meteorological variable in the troposphere in MOCAGE and set from the forcing meteorological model. In most previous modelling studies, $H_2S$ and CO volcanic emissions have not been included.. In Table 1, $H_2S$ emissions are much smaller than $SO_2$ emissions. We checked a posteriori that $H_2S$ and CO emissions are not

important, making negligible changes in the model results with or without their inclusion.

The values given in Table 2 serve for the reference simulations called "N.Ref" for the eruption. To trigger fast initial production of BrO, Br emissions are included as in Surl et al. (2021). Surl et al (2021) concluded that emitted OH's main effect on bromine chemistry was to produce Br radicals from HBr shortly after emission. In the absence of primary Br emissions, differing quantities of OH had an effect that had largely dissipated by 30 minutes after emission. Since our work aims at preparing 3D

simulations at the regional and global scales at least over several hours, and includes primary Br emissions, it is possible to neglect OH emission. Concerning NOx that may also be produced by high temperatures, as explained before, it is not included in emissions in the reference simulations. However, additional experiments are done to test the sensitivity to the composition of the volcanic emissions, in particular including NOx emissions, as detailed in Section 4.3.

**Table 2: Emissions used as input for MOCAGE 1D model for the reference experiment N.Ref (see explanations in the text).**

| Species | Molar ratio to $SO_2$ | Eruption emissions in tons between 14.15 and 18.15 UTC |
|---|---|---|
| $SO_2$ | 1 | $8.00\ 10^3$ |
| HCl | 0.3 | $1.37\ 10^3$ |
| $H_2S$ | $6.6\ 10^{-3}$ | $2.70\ 10^1$ |
| CO | $3.1\ 10^{-3}$ | $1.09\ 10^1$ |
| HBr | $2.46\ 10^{-4}$ | 2.50 |
| Br | $0.82\ 10^{-4}$ | $8.21\ 10^{-1}$ |
| Primary sulphate aerosols | 0.02 | $2.40\ 10^2$ |

The end time of the eruption (18.15 UTC) is very close to night time. Thus, the role played by photochemistry in the plume can only be fully analysed when daylight comes back the day after (11 May in the morning with dawn daylight starting at 4:15 UTC). This is why we have also set another experiment called D.Ref that is the same as N.Ref except that the 4h eruption

occurs at the start of daytime from 04.15 UTC on 11 May instead of 14.15 UTC on 10 May, so that the bromine cycle is not stopped early by night time conditions. The chemical initial conditions for these daytime simulations are from the same MOCAGE-3D simulation as for N.Ref but on 11 May at 04.00 UTC. The daytime simulations do not represent the real eruption but are of interest for studying the bromine cycle in daylight conditions, conditions which are most favourable to the bromine cycle. These simulations are run until 11 May 18 UTC, just before night. Note that the simulations run with the eruption





stopping at 18:15 UTC and including night-time conditions are referenced as 'N.' and those run over only daytime with the eruption stopping at 08:15 UTC, are referenced as 'D.'.

Another parameter that needs to be set in the simulations is the effective radius of the sulphate aerosols ($R_{eff}$) corresponding to the mean surface area-weighted radius. It is used to calculate the total surface of sulphate aerosols which is one of the parameters of the heterogeneous reaction rate constants (reactions (R5a), (R5b) and (R6)). A few studies give an estimate of

the value of the sulphate aerosol radii within Mount Etna plumes close to the vent (Watson and Oppenheimer, 2000, 2001; and Spinetti and Buongiorno, 2007, Roberts et al. 2018). Watson and Oppenheimer (2000; 2001) measured a mean effective radius of ~0.7 to 0.85 μm in M$^t$ Etna's plumes. Spinetti and Buongiorno (2007) airborne observations of M$^t$ Etna's plumes gave $R_{eff}$ = ~1 μm. More recently, Roberts et al. (2018) found $R_{eff}$ = 0.3 μm from measurements of aerosol size distributions gathered in passive emissions of M$^t$ Etna. $R_{eff}$ is expected to vary depending on the environmental conditions and the characteristics of the

emission. The differences between these studies may also come from limitations of the aerosol observations used, in particular regarding the sampling of small particles that can be underestimated. This is why we choose here $R_{eff}$ = 0.3 μm from Roberts et al. (2018) since this value was inferred from ash-free observations in quiescent degassing over a wide range of aerosol sizes, including small size particles.

The reference simulations are listed in Table 3. Additionally, the 'N.BGD' (resp. 'D.BDG') simulation is run with no volcanic

emissions in night (resp. daytime) conditions to characterise the background chemical conditions for appropriate species.

**Table 3: List of the reference simulations described in Section 4.1 and of the simulations using the subgrid scale plume parameterization described in Section 4.2.**

| Simulation name | Night/Day | Eruptive emissions | Plume parameterization | X value if plume parameterization |
|---|---|---|---|---|
| N.Ref | Night | Yes | No | N/A |
| D.Ref | Day | Yes | No | N/A |
| N.Plume.0.3 | Night | Yes | Yes | 0.3 |
| N.Plume.0.1 | Night | Yes | Yes | 0.1 |
| D.Plume.0.3 | Day | Yes | Yes | 0.3 |
| D.Plume.0.1 | Day | Yes | Yes | 0.1 |
| N.BGD | Night | No | N/A | N/A |
| D.BGD | Day | No | N/A | N/A |

**4.2 Subgrid-scale parameterization**

The study is focused on a 1-D model, but using the characteristics of the 3-D MOCAGE model. 3-D chemical models resolve the chemical reactions at the grid box scale with the assumption that chemical species are homogeneously distributed within



each grid box. However, within a volcanic plume, the bromine chemistry takes place within a smaller volume compared to the usual volume of MOCAGE grid boxes: from $2° \times 2°$ to $0.5° \times 0.5°$ for global simulations and from $0.5° \times 0.5°$ to $0.1° \times 0.1°$ for

regional simulations. Thus, at the grid scale of global models, volcanic eruption plumes can be considered as a sub-grid scale phenomenon. Processes occurring at sub-grid scales are typically represented via parameterizations in atmospheric models. For atmospheric plume modelling, the Plume-in-Grid approach is the most widely used (see review by Karamchandani et al., 2011), in particular for air quality applications, giving a better description of the phenomenon studied. The principle of the Plume-in-Grid is to use a reactive plume model in addition to the 3-D model. This reactive plume model is a representation of

three-dimensional puffs. Using this method requires running the reactive plume model in addition to the 3-D model. We propose here a simple version of the Plume-in-Grid approach designed to be not too computationally expensive for possible further use in a 3-D global/regional CTM framework and long-term simulations. The basis of this parameterization is to represent the subgrid-scale chemical reactions at the plume scale only in the model vertical column in which the volcano is located, as in Grellier et al. (2014). It consists of computing the chemical reactions defined by the model within a volume of

$0.025° \times 0.025° \times$ height (~2.5 km x ~2.5 km x height) of the gridbox (called hereafter Plume box) representative of the plume area, which is much smaller than the model grid volume (called hereafter Model box). Therefore, the ratio of the volume of the Plume box over the Model box equals 1/400. We also define the Model-P box which is the Model box minus the volume of the Plume box (volume Model-P box/volume Model box = 399/400). This parameterization has three steps:

- During the time of the emission, at every model timestep (15 min), the first step is to include the volcanic emission
fluxes within the Plume box.

- The second step is to calculate separately the chemical productions and losses within the Model-P box and within the Plume box.

- In the third step, the volume fluxes of the chemical species calculated within the Plume box are mixed with the species within the Model-P box, whose concentrations were calculated separately in the second step.

Grellier et al. (2014) proposed two possibilities for the computation of the third step. One was to add the full content of the Plume box to the Model-P box at each model time step (called "Plume 1"). In this case, we assumed that the plume undergoes complete mixing with the model grid box every 15 min. The second possibility was to add the Plume box content with the Model-P box only at the end of the eruption (called "Plume 2"). This means steps 1 and 2 are calculated in parallel throughout the eruption without interaction and the third step is only undertaken at the end of the 4-hour emission period. These two

approaches correspond to two extreme assumptions for the dilution of the plume but neither of them is realistic. This is why we developed an intermediate and more realistic approach with a partial mixing at each model timestep of the Plume box concentrations with the Model-P box at a rate X, with X ranging between 0 and 1 ("Plume.X" simulations). A low value of X corresponds to a low dilution rate of the Plume box with the Model-P box at each timestep (15 min). Plume 1 and Plume 2 configurations from Grellier et al. (2014) correspond to X=1 and X=0, respectively. We also assume that the plume dilution

takes place continuously even after the end of the eruption with the same X rate. The detailed description of the parameterization is given in Fig. 1.





X represents the rate of dilution of the Plume box of size ~2.5 km x 2.5 km with the Model-P box a size of ~50 km x 50 km. In reality, it varies with the plume characteristics and the meteorological conditions. This is why we test two values of X here : X=0.3 and X=0.1. This corresponds to a sensible range for full dilution time of ~2.5 hours for X=0.3 and ~10 hours for X=0.1

after the end of the eruption, respectively.

The simulations including the subgrid scale plume parameterization are listed in Table 3.

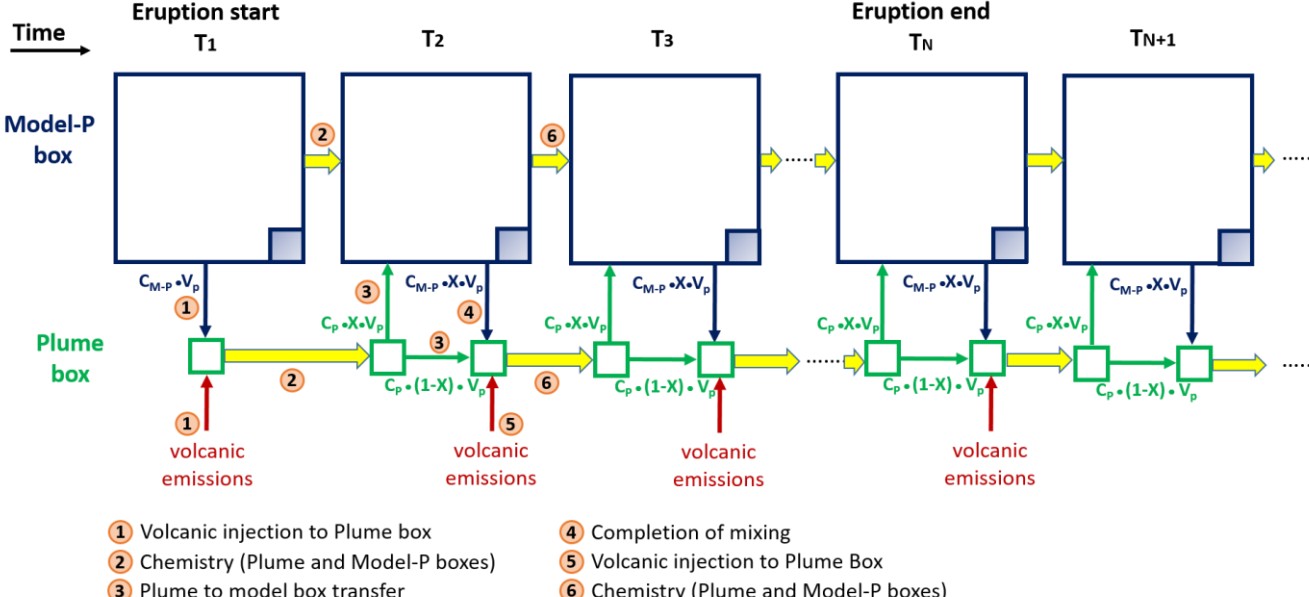


**Figure 1: Schematic representation of the subgrid scale parameterization of plume chemistry. At the timestep corresponding to the start of the eruption the Plume box and the Model-P box are defined: Plume box has a volume ($V_P$) of 0.025° x 0.025° x height and Model-P Box is defined as the Model box (0.5° x 0.5° x height) minus $V_P$ (shaded blue square minus the big blue square) which volume is noted $V_{M-P}$. The concentrations in the Plume box and Model-P boxes are noted $C_P$ and $C_{M-P}$, respectively. (Step 1) At $T_1$,**

**the Plume box initial chemical concentrations are the background concentrations from the Model-P box are added to the volcanic emissions over the timestep (15 min). (Step 2) The chemistry is applied separately to the Plume Box and to the Model-P Box (big arrows in yellow colour). (Step 3) At $T_2$, a fraction X of the molecules contained in the Plume box $C_P \bullet (X \bullet V_P)$ are transferred to the Model-P box and $(1-X) \bullet V_p$ kept in the Plume box. (Step 4). To complete the mixing between Plume and Model-P boxes, the Model-P box transfers $C_{M-P} \bullet X \bullet V_p$ to the Plume box. This is at this step that the concentrations in the Model box are output from adding**

**Model-P+Plume concentrations. (Step 5) The concentration of the Plume box is then updated by adding the volcanic emissions. (Step 6) The chemistry is applied separately to the Plume Box and to the Model-P Box. (big arrows in yellow colour). For the subsequent model timesteps until the end of the eruption, steps 3 to 6 are repeated. After the end of the eruption, Steps 3, 4 and 6 are repeated, i.e. excluding the step of volcanic emissions.**





### 4.3 Sensitivity tests

Several sensitivity tests were performed regarding the emission amount and composition and the primary sulphate characteristics. These simulations are only run in the daytime configuration in order to best follow the bromine explosion since the night configuration stops BrO production very rapidly just after the end of the eruption. Also because daytime simulations are shorter (14 hours) and the maximum of BrO is reached not long after the end of the eruption (e.g., < 2 hours in D.Ref simulation), there is less of an expected effect arising from the assumption of no exchange between the selected 1D column and its surrounding background air. These sensitivity simulations are performed with the sub-grid scale parameterization only when relevant.

Roberts et al (2014) and Roberts (2018) showed that the relative production of BrO from HBr depends on the emission flux and on the total bromine (HBr+Br) /$SO_2$ ratio of the emissions. This is why sensitivity simulations are run with lower emission fluxes for all species and with a lower total bromine/$SO_2$ ratio including also the subgrid-scale parameterization since the bromine partition may depend in these cases on the size of the box considered and associated concentrations. Their characteristics are given in Table 4.

**Table 4: Characteristics of the test simulations on the amount of emissions and on the total bromine/$SO_2$ ratio.**

| Simulation name | Plume parametrization | X value (plume param.) | $SO_2$ eruption emissions in tons between 04.15 and 08.15 UTC | Total bromine/$SO_2$ molar ratio |
|---|---|---|---|---|
| D.LowEmis | No | N/A | $8.00 \times 10^2$ | $3.28 \times 10^{-4}$ (as in D.Ref) |
| D.LowEmis.Plume.0.3 | Yes | 0.3 | $8.00 \times 10^2$ | $3.28 \times 10^{-4}$ (as in D.Ref) |
| D.LowEmis.Plume.0.1 | Yes | 0.1 | $8.00 \times 10^2$ | $3.28 \times 10^{-4}$ (as in D.Ref) |
| D.LowHBr | No | N/A | $8.00 \times 10^3$ (as in D.Ref) | $3.28 \times 10^{-5}$ |
| D.LowHBr.Plume.0.3 | Yes | 0.3 | $8.00 \times 10^3$ (as in D.Ref) | $3.28 \times 10^{-5}$ |
| D.LowHBr.Plume.0.1 | Yes | 0.1 | $8.00 \times 10^3$ (as in D.Ref) | $3.28 \times 10^{-5}$ |

Other sensitivity simulations are listed in Table 5. Firstly, we analyse the sensitivity of the rapid formation of BrO to the composition of the volcanic emissions, in particular to assess the individual impact of the additional species produced at the vent and by high temperature processes (Br and primary sulphate). We also run simulations with different Br/HBr and primary sulphate/$SO_2$ ratios since these ratios vary naturally with the characteristics of the volcano's emissions and their environmental conditions and also because their determination is still uncertain as discussed in previous sections. In addition to Br emissions, we also test the inclusion of oxidants in the form of NOx that are possibly formed at high temperatures. For this, we use the NO/$SO_2$ molar ratio of $4.5 \times 10^{-4}$ of Surl et al. (2021).

Another important parameter that drives the bromine explosion is the total surface area of sulphate aerosols for the heterogeneous reactions. It is calculated in our simulations from the sulfate concentration and the $R_{eff}$ parameter. Because of the natural variations of $R_{eff}$ and the uncertainties on its estimation from observations, we performed sensitivity tests with other $R_{eff}$ values based on estimates of $R_{eff}$ from previous studies (see section 4.1): $R_{eff} = 0.7$ μm and $R_{eff} = 1.0$ μm, instead of $R_{eff} =$



0.3 μm in the reference simulations. Apart from $R_{eff}$, their settings are the same as in the D.Ref simulation. Thus, for a given sulfate concentration, a higher $R_{eff}$ leads to a fewer number of larger particles and a lower aerosol surface area.

Among the parameters that may not be well observed, there is also the top altitude of the eruption. For volcanoes located in remote places, this altitude can be estimated from satellite observations but with uncertainties (e.g. Scollo et al. 2014; Corradini et al. 2020). This is why we test here the influence of the top height of the eruption, ranging from 7.5 to 9.5 km.

**Table 5: Characteristics of the other test simulations described in Section 4.3.**

| Simulation name | Primary sulphate emission | Br emission (% of HBr) | NO emission (NO/SO$_2$ molar ratio) | $R_{eff}$ | Eruption top altitude |
|---|---|---|---|---|---|
| D.Emis.NoHT | No | No | No | 0.3 μm | 8.5 km |
| D. Emis.NoSulf | No | Yes (25%) | No | 0.3 μm | 8.5 km |
| D.Emis.Sulf2 | Yes (4%) | Yes (25%) | No | 0.3 μm | 8.5 km |
| D.Emis.NoBr | Yes (2%) | No | No | 0.3 μm | 8.5 km |
| D.Emis.Br50 | Yes (2%) | Yes (50%) | No | 0.3 μm | 8.5 km |
| D.Emis.NO | Yes (2%) | Yes (25%) | Yes (4.5 10$^{-4}$) | 0.3 μm | 8.5 km |
| D.Reff.0.7 | Yes (2%) | Yes (25%) | No | 0.7 μm | 8.5 km |
| D.Reff.1.0 | Yes (2%) | Yes (25%) | No | 1.0 μm | 8.5 km |
| D.Alt.9.5 | Yes (2%) | Yes (25%) | No | 0.3 μm | 9.5 km |
| D.Alt.7.5 | Yes (2%) | Yes (25%) | No | 0.3 μm | 7.5 km |

## 5 Results

All the results shown in this section are partial column concentrations vertically integrated over the volcanic emission levels in molecules.cm$^{-2}$, i.e., from 3300m to 8500m in all simulations except for the sensitivity simulations to the top altitude of the eruption for which the top altitude is 7500m or 9500m, instead of 8500m. The figures show the concentrations in the Model box. For the Plume.0.3 and Plume.0.1 simulations, the concentrations in the Model box come from adding the Model-P box and the Plume box concentrations at each 15 min timestep. Note that because the eruption starts at 14:15 UTC for N.

simulations (resp. 04:15 UTC for D. simulations) and that the main time step of the model is 15 minutes, the effect of the emissions is only visible in the figures at 14:30 UTC (resp. 04:30 UTC for D. simulations).

### 5.1 Analysis of the reference and plume parameterization simulations for the eruption starting in the afternoon

The time evolution of the column of BrO, HBr, O$_3$, NOx, OH and the ratio BrO/SO$_2$ (in red) for N.Ref simulation is shown in Fig. 2. Additionally, the partitioning between the bromine species for N.Ref is shown in Fig. 3a. BrO (Fig. 2a) formation is

triggered just after the start of the eruption and increases rapidly until 17:45 UTC. During this period, HBr is efficiently converted into BrO (up to 55%) and a small part into HOBr (5%) (Fig. 3a). The Br contribution to the total bromine is very low because of its very rapid conversion to BrO. BrONO$_2$ has only a small contribution to the total bromine because it is efficiently depleted by hydrolysis. After 17:45 UTC, when solar radiation decreases and therefore photolysis becomes less efficient, BrO decreases rapidly.


**Figure 2: Time evolution from 14:15 UTC of the column number of molecules of BrO (a), HBr (b), O₃ (c), NOx (d) and OH (e) by unit surface and of the ratio BrO/SO₂ (f) within the Model box (Model-P box + Plume box) for the N.Ref, N.Plume.0.3 and N.Plume.0.1, and N.BGD where appropriate. The quantities are integrated vertically on the emission levels (3300m-8500m). The green zone corresponds to the 4 hours of the volcanic eruption emission (14:15-18:15 UTC) and the light grey zone to night-time.**



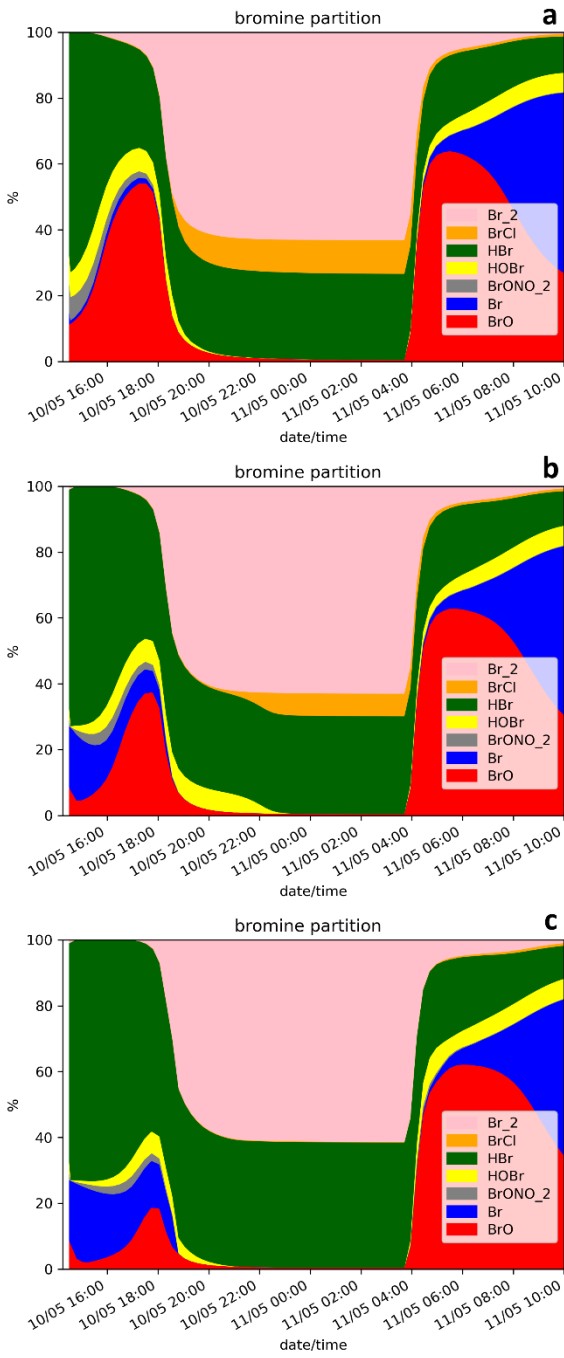

**Figure 3: Time evolution from 14:15 UTC of the relative partition of the bromine species in % for N.Ref and (a), for N.Plume.0.3 (b) and for N.Plume.0.1 (c) simulations within the Model box. The partition is calculated from total bromine $Br_y$ with $Br_y$ = HBr + BrO+ Br + 2 $Br_2$ + BrCl + HOBr + BrONO$_2$.**





At night-time BrO disappears after about one hour to produce reservoir species, mainly $Br_2$ and to a lesser extent BrCl. $Br_2$ production is dominant (reaction (R5a)) with respect to BrCl (Reaction (R5b)) at high HBr/HCl ratio. But because HBr is largely depleted before night, the HBr/HCl ratio becomes small enough to lead to some production of BrCl. The fraction of HBr remaining from the emission and not converted into BrO before sunset is stable during the night because of the lack of photolysis.

On the day after the eruption (11 May) upon sunrise, the bromine cycle starts again, using HBr to produce BrO rapidly until ~5.30 UTC (max ~6.5 $10^{14}$ molecules.cm$^{-2}$). After ~5.30 UTC, the contribution of Br increases while BrO decreases (Fig. 3) because of decreasing concentrations of $O_3$ (Fig. 2d). Such an increase of Br was also found in the model results at the regional scale of Jourdain et al. (2016) from 70 km downwind from the Ambrym vent. Here the Br enhancement is expected to be stronger than in Jourdain et al. (2016) towards the end of the simulation since we assume no mixing with air outside the 1D column leading to a lack of oxidants that could come from background surrounding air. Figure 2c shows that the halogen (chlorine and bromine) cycling, including the production of BrO, depletes ozone significantly in the plume during daytime on the day of the eruption and even further on the day after, leading to about half of the initial ozone at the end of the N.Ref simulation.

During daytime on the day of the eruption and on the day after, the bromine cycle leads to an $O_3$ decrease (Fig. 2c) together with NOx (Fig. 2d), OH (Fig. 2e) depletion and $HNO_3$ formation (not shown). Overall, the results of the N.Ref simulation (bromine partition and depletion of oxidants) are consistent with previous modelling studies of bromine in volcanic plumes (e.g., Roberts et al. 2009, Jourdain et al. 2016, Surl et al 2021).

To characterise the efficiency of the bromine cycle in the plume compared to observations, we calculate the BrO/SO$_2$ ratio. The simulated values of BrO/SO$_2$ are well within the range of variation of observed BrO/SO$_2$ ratio at Mt Etna (Gutmann et al. (2018). In the simulation, the time variation of BrO/SO$_2$ ratio (Fig. 2f) is similar to BrO (Fig. 2a). Note that at 14:30 UTC, the first timestep when the emissions are injected and have been processed chemically, BrO/SO$_2$ shows a stronger gradient compared to the next timesteps. This is because the 14:30 UTC timestep benefits from high background OH concentrations that are largely used to produce BrO from Reactions (R1) and (R2), in addition to the production of BrO through heterogeneous reactions. At later timesteps, there is less OH leading to less steep variations of BrO.

To evaluate if the simulation gives reasonable estimates, BrO and SO$_2$ columns are compared to satellite retrievals from the GOME-2 space-borne instrument. In the supplement of Hörmann et al. (2013), observations of BrO and SO$_2$ in different volcanic plumes are presented, in particular those of the Mount Etna plume on 11 May, originating from the 10 May eruption. The overflight time of GOME-2 above the volcanic plume was 08:40 UTC. The values of the observations and of the simulation are both integrated columns in molecules cm$^{-2}$. For GOME-2, the data correspond to slant column densities, and are therefore slightly different from the model-derived columns. Note that the model results are the partial columns over the emission levels but they can be assimilated to tropospheric columns since background SO$_2$ concentrations are by far lower than those from the eruption and bromine species are initialised to zero in the troposphere. At the time of the overpass of GOME-2, the observed BrO maximum and the corresponding SO$_2$ value are $2.3\times10^{14}$ and $1.6\times10^{18}$ molecules cm$^{-2}$, respectively. In the N.Ref





simulation at the same time, BrO and $SO_2$ columns are $3.5 \times 10^{14}$ and $3.1 \times 10^{18}$ molecules $cm^{-2}$, respectively. The observed and simulated values of BrO and $SO_2$ are reasonably close, although higher in the simulations. There are several explanations for this. The absence of transport and deposition in the 1-D MOCAGE simulations leads to no dilution of the plume or loss by deposition and thus to an expected overestimation compared to observations. Moreover, the concentrations of the chemical species are representative of a larger surface area in the observations compared to the model grid size, the satellite pixel being $40 \times 80$ $km^2$ and the model grid box being $0.5° \times 0.5°$ ($\sim 44 \times \sim 55$ $km^2$). Overall, the agreement is very good, considering also the uncertainties of the satellite estimates of $SO_2$ and BrO columns due to instrument sensitivity, atmospheric conditions and the assumptions used in the retrieval method, and of the estimates of the volcanic gas fluxes and their composition used in MOCAGE. An additional and pertinent way to evaluate the simulation is to compare $BrO/SO_2$ ratios. In the supplement of Hörmann et al. (2013), the ratio of integrated molecules within the plume ($1.24 +/- 0.19$) $\times 10^{-4}$ can be compared to the N.Ref $BrO/SO_2$ column ratio at the time of the observation ($1.13 \times 10^{-4}$). The model $BrO/SO_2$ value is consistent with that derived from the GOME-2 observations showing a realistic production of BrO in the model.

Figure 2 also shows the time evolution of the species for N.Plume.0.3 and N.Plume.0.1 simulations, simulations using the plume parameterization. The partition of the bromine species for N.Plume.0.3 and N.Plume.0.1 is depicted in Fig. 3. N.Plume.0.3 and N.Plume.0.1 have similar overall variations as N.Ref. However, during the eruption, when applying the plume parameterization, BrO maximum values (Fig. 2a) are lower and corresponding to higher values of HBr (Fig. 2b) and ozone (Fig. 2c). This is due to the number of molecules of oxidants ($O_3$, HOx, NOx) available in the Plume Box that is lower than in the Model box because of the volume difference. This limits the bromine explosion cycle in the Plume box and therefore BrO production. This is consistent with the bromine partitioning shown in Fig. 3 that with less dilution between the Plume box and the Model-P box (from X=0.3 in N.Plume.0.3 to X=0.1 in N.Plume.0.1), less Br is converted into BrO. This behaviour of the plume parameterization, whereby mixing controls the production of BrO by limiting the availability of oxidant, is comparable to the studies of the observed behaviour and modelling results within the core volcanic plumes (e.g., Bobrowski et al., 2006; Jourdain et al., 2016; Roberts 2018, Surl et al.; 2021). Such studies show that BrO production is limited by the amount of oxidants available, and that BrO production is higher at the edge of plumes where there is mixing with oxidant-rich background air compared to within the plume core. Ozone (Fig.2c), NOx (Fig. 2d) and OH (Fig. 2e) depletion occurs during the eruption thanks to BrO net formation but this gets less strong in the Model-P box as the dilution rate decreases. The time variation of $BrO/SO_2$ ratio (Fig. 2f) is consistent with that of BrO (Fig. 2a). The main difference is visible from the first timesteps of the volcanic emission. At 14:30 when the emission is first taken into account, there is an increase of $BrO/SO_2$ in N.Plume.0.3 and N.Plume.0.1 because the Plume box was initialised with background concentrations providing enough oxidants to produce BrO. In the timestep after (14:45 UTC), there are less oxidants available in the Plume box and yet not very much of the emissions transferred to the Model-P box to produce BrO efficiently in the Model-P box. From 15:00 UTC, $BrO/SO_2$ increases mainly because of BrO production in the Model-P box from the partial mixing with the emission-rich Plume box at each time step. With low X, this mixing is slow leading to a less steep $BrO/SO_2$ ratio increase. The discontinuity in the first timestep of the eruption is likely not realistic. This is only due to the emissions being injected every 15 minutes (model timestep) and due





to the initialisation of the Plume box. Nevertheless, after these few first timesteps, the behaviour of MOCAGE-1D $BrO/SO_2$ with the plume parameterization shows results consistent with previous studies and with N.Ref simulation.

At night, the partition of the bromine species is different in the 3 simulations (Fig. 3). In N.Plume.0.1, the reservoir species at night is $Br_2$ only since the HBr/HCl ratio is such that reaction (R5b) is not active. N.Plume.0.3 simulations show an intermediate situation which favours firstly $Br_2$ production until the HBr/HCl ratio is sufficient to trigger the production of BrCl via reaction (R5b). As in N.Ref, $Br_2$ and BrCl concentrations are stable in time once HOBr is fully depleted.

On daytime on 11 May (from 4:15 UTC), the maxima of BrO and $BrO/SO_2$ ratio in N.Plume.0.3 and N.Plume.0.1 simulations reach values close to the N.Ref simulation, but are slightly lower and occur a bit later as X (dilution rate) decreases from 0.3 to 0.1. The BrO and HBr concentrations tend to converge for all simulations on 11 May from 5 UTC consistently with the partitioning between the bromine species that is very similar in the three simulations (Fig. 3). This is because the day after the eruption the emissions injected in the Plume box had enough time to be fully diluted in the Model-P box. $O_3$ strongly decreases from 11 May 04:15 UTC in N.Plume.0.3 and N.Plume.0.1 as in N.Ref with small differences for N.Plume.0.1 at the end of the simulation linked to a slightly lower net production of BrO. As for N.Ref, results from the plume parameterization simulations for $BrO/SO_2$ ratio at the time of the observations are within the GOME-2 estimated range ($(1.24 +/- 0.19) \times 10^{-4}$) with 1.15 and 1.17 for N.Plume.0.3 and N.Plume.0.1, respectively. These ratios are not very different from N.Ref because of a similar behaviour of the Plume simulations the day after the eruption.

In summary, the MOCAGE-1D simulations provide results consistent with observations and with previous modelling studies. This means that the MOCAGE-1D model, with the update to the MOCAGE chemical scheme to account for the halogen plume chemistry is able to simulate well the bromine cycle in volcanic plumes. The use of a sub-grid scale plume parameterization changes the results mainly during the eruption. During the eruption, the parameterization reduces BrO net production similarly to what occurs in the core of the plume because of less oxidants being available. The results of the simulations with and without the plume parameterization converge on the day after the eruption because most of the plume emissions are already diluted in the model grid box giving a similar efficiency of HBr conversion to reactive bromine. The effect of the parameterization is only to slightly reduce and delay the $BrO/SO_2$ maximum.

## 5.2 Analysis of the reference and plume parameterization simulations for the eruption starting early morning

Since the night starts just after the end of the eruption and photolysis plays a role in the bromine cycle, we also study the impact of the subgrid-scale parameterization assuming an identical eruption emission but that takes place during daytime, from 04:15 UTC on 11 May instead of 14:15 UTC on 10 May. The results for D.Ref, D.Plume.0.3 and D.Plume.0.1 are displayed in Figs. 4 and 5. In Fig. 4, D.Ref simulation shows a rapid increase of BrO and $BrO/SO_2$ with time from HBr emissions leading to strong decreases in ozone (Fig. 4c), NOx (Fig. 4d) and OH (Fig. 4e) compared to background (D.BGD). BrO reaches a maximum of $5.7 \times 10^{14}$ molecules $cm^{-2}$ at ~9 UTC which is a bit lower than in N.Ref (eruption at the end of the day) on the day after the eruption ($6.3 \times 10^{14}$ molecules $cm^{-2}$). This is explained by higher initial ozone concentrations on 10 May 14:15 UTC.





**Figure 4: Similar to Fig. 2 but for the daytime simulations D.Ref, D.Plume.0.3 and D.Plume.0.1 from 04:15 UTC. The green zone corresponds to the 4 hours of the volcanic eruption emission (4:15-8:15 UTC).**





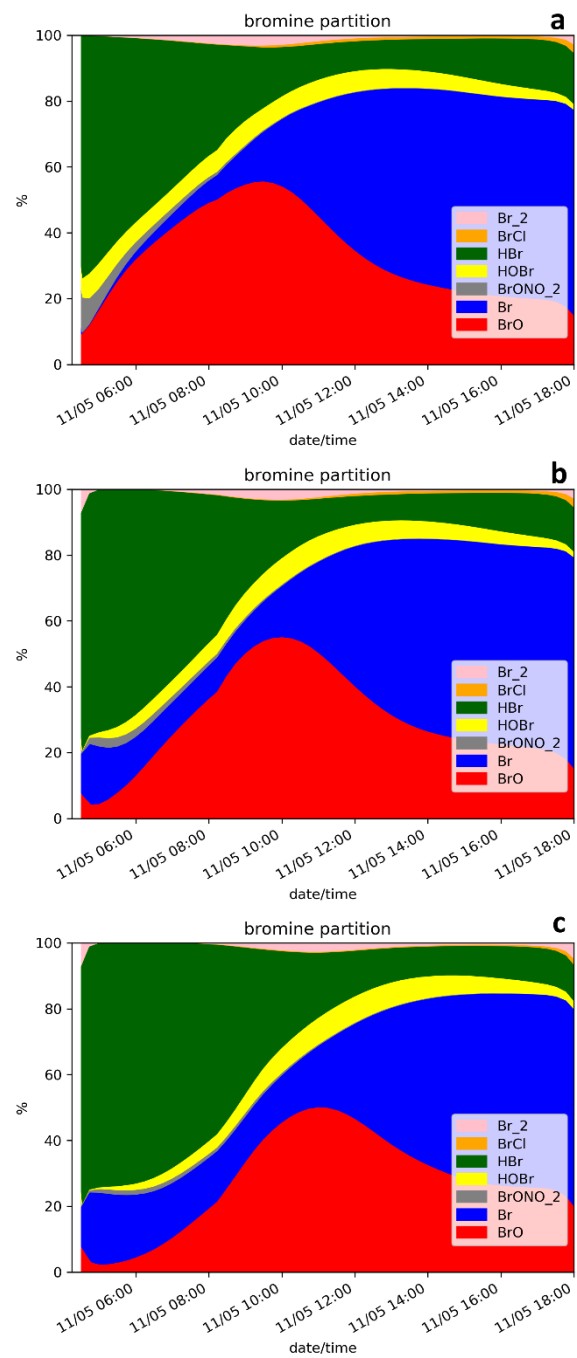

**Figure 5: Similar to Fig. 3 but for the daytime simulations D.Ref (a), D.Plume.0.3 (b) and D.Plume.0.1 (c) from 04:15 UTC.**

When applying the plume parameterization, the BrO maximum tends to be lower and to occur later with decreasing dilution rate. For X=0.3 (D.Plume.0.3), at the time when BrO is at a maximum, ~95 % of the Plume box is already mixed with the



Model box. This is why the results of D.Ref and D.Plume.0.3 are similar with a maximum of ~55% of BrO in the partitioning of bromine species (Figs. 5a and 5b). For X=0.1, the maximum is lower (~5.0 $10^{14}$ molecules $cm^{-2}$) and occurs at 11 UTC corresponding to ~48% of the bromine species (Fig. 5c). The lower dilution rate slows down the production of BrO since less molecules of oxidant species are available in the Plume box compared to the Model-P box. This leads to more HBr and more ozone remaining in Model Box (Model-P +Plume box) D.Plume.0.1 simulation. The differences between the plume simulations are higher in the D. than in the N. simulations. On the day after the eruption, BrO production in N. simulations is quicker because part of the HBr is transformed into the form of $Br_2$ and BrCl that are rapidly photolysed and converted to BrO thanks to higher ozone and because the full mixing between Plume and Model-P box is already mostly reached, even in N.Plume.0.1. Note that at the very end of the D. simulation the rapid decrease of BrO is due to nightfall. $BrO/SO_2$ ratio (Fig. 4f) follows mostly BrO variations except for the first timesteps of the emissions with the same behaviour as discussed for the N. simulations (section 5.1).

The bromine partitioning (Fig. 5) shows similarities between the D.Ref, D.Plume.0.3 and Plume.0.1 simulations. However, before the BrO maximum is reached, as the dilution rate decreases, there is more Br and less BrO. This behaviour is consistent with the results of the N. simulations (Fig. 3) before night-time. This is explained by ozone being quickly depleted in the Plume box, slowing down the overall BrO production in the Model box (Model-P+Plume box). After the maximum of BrO is reached, Br increases while BrO decreases. Similarly to the N. simulations, this is due to decreasing concentrations of $O_3$.

In summary, MOCAGE-1D early morning eruption experiments simulate the bromine cycle in a consistent way, and for D.Ref a maximum in BrO is reached about 1.5 hour after the end of the eruption. As for the N. simulations, the sub-grid plume parameterization slightly delays BrO formation and the maximum BrO reached is lower as the dilution rate decreases. There is an overall good consistency between the N. and D. simulations with differences due to the night interrupting BrO formation, higher initial ozone levels and more dilution of the plume in the model box at the time of the BrO maximum because it occurs later in the N. simulations.

## 5.3 Other sensitivity tests

Hereafter, we only discuss the sensitivity simulations starting on 11 May 04:00UTC. Such D. simulations are consistent with N. simulations but the bromine cycle is not interrupted by night. This leads to a shorter time to reach the BrO maximum in the D. configuration, meaning that there is less impact of the 1D-framework limitation linked to the assumption of no mixing with air surrounding the 1D model profile. It was checked that the results of the sensitivity simulations in N. configuration are consistent with those in D. configuration. The characteristics of the sensitivity tests discussed in this section are given in Tables 4 and 5. For the simulations not including the plume parameterization (Table 5), it was checked that running MOCAGE-1D with both the plume parameterization and the sensitivity parameters does not change the conclusions of the sensitivity analysis.





### 5.3.1 Sensitivity to the total bromine/SO₂ emission ratio and to the emission flux

The bromine partitioning from the simulations with a lower total bromine/SO$_2$ emission ratio or a lower emission flux (Table 4) are shown in Fig. 6, including also the plume parameterizations (simulations D.LowHBr, D.LowHBr.Plume.0.3 and D.LowHBr.Plume.0.1, and D.LowEmis.Plume0.3, D.LowEmis.Plume0.1).

Bromine partitioning in these simulations differs from the reference simulation D.Ref (Fig. 5). There is a much stronger decrease of HBr compared to D.Ref, and consequently there is more formation of BrCl (Reactions (5a) and (5b)). There is a stronger and more sustained increase in BrO, whilst the proportion of Br is lower than in D.Ref. Another difference lies in HOBr (and BrONO$_2$) which is proportionally higher in D.LowHBr and D.LowEmis than in D.Ref. These differences are related to the degree of mixing of oxidants relative to halogens in the plume (i.e. with relatively more oxidants for an emission with lower total bromine/SO$_2$ or lower emission flux). The results are consistent with the 1D model sensitivity studies on gas flux and plume-air mixing of Roberts et al. (2014), although the time-evolution of bromine speciation for this strong eruptive emission differs to the passive degassing case. The impact of the plume parameterization when the total bromine/SO$_2$ emission ratio is low (D.LowHBr.Plume.0.3 and D.LowHBr.Plume.0.1 and and D.LowEmis.Plume0.3, D.LowEmis.Plume0.1) is to cause an initial enhancement in the proportion of BrO in the first timesteps of the eruption compared to D.LowHBr, while this effect was not seen in D.Plume.0.3 and D.Plume.0.1 compared to D.Ref. In the case of a low total bromine/SO$_2$ ratio or emission flux, the composition in the Plume box is such that there are sufficient oxidants (ozone, NOx, HOx) to produce BrO efficiently from HBr compared to D.Plume.0.3 and D.Plume.0.1. However, during these first timesteps, these oxidants are rapidly consumed in the plume parameterizations cases, leading thereafter to a decrease in BrO with respect to HBr, more evident in D.LowHBr.Plume.0.1 and D.LowEmis.Plume0.1 because of its lower dilution rate. Over the duration of the simulation after the eruption injection, the higher oxidant to halogen ratio in these simulations compared to the reference leads to enhanced HOBr (formed from reaction of BrO with HO$_2$) and lower Br (removed by reaction of Br with O$_3$ to form BrO). There are some subtle differences between the simulations with lower emission flux (D.LowEmis, D.LowEmis.Plume.0.3 and D.LowEmis.Plume.0.1) and lower total bromine/SO$_2$ emission ratio (D.LowHBr, D.LowHBr.Plume.0.3 and D.LowHBr.Plume.0.1): in the case of lower emission flux, the decrease in HBr is somewhat slower, and the proportion of HOBr is greater. This can be understood in terms of less aerosol surface available for heterogeneous chemistry in the lower emission flux case, resulting in slower conversion of HBr into reactive halogens and a smaller sink for HOBr. Overall, this sensitivity study emphasizes the complex interplay between halogens, oxidants and aerosol in the formation and partitioning of reactive halogen species including BrO, through chemical reactions that in turn deplete the atmospheric oxidants.



**Figure 6: Similar to Fig. 5 but for the simulations D.LowHBr (a), D.Low.Emis (b), D.LowHBr.Plume.0.3 (c), D.LowEmis.Plume.0.3 (d), D.LowHBr.Plume.0.1 (e) and D.Low.Emis.Plume.0.1 (e).**



**Figure 7: Similar to Fig. 4 but for the simulations testing the sensitivity to emission composition (in particular the primary sulfate and high-temperature products): D.Ref, D.Emis.NoHT, D.Emis.NoSulf, D.Emis.Sulf, D.Emis.NoBr, D.Emis.Br50, D.Emis.NO and D.BGD when appropriate. Details on these simulations are given in Table 5.**





### 5.3.2 Sensitivity to the emission composition from high temperature processes

We analyse here the sensitivity of the bromine cycle to variations in the composition of the emissions resulting from high temperature processes (see Table 5). The results of these tests are depicted in Fig. 7.

D.Emis.NoHT simulation corresponds to the use of the raw emissions from Table 1, meaning not accounting for the change of composition at high temperature when magmatic gases first encounter atmospheric air. We know that this simulation is not

realistic but it gives the lower bound of BrO production since it corresponds to emissions without Br radicals and primary sulphate. D.Emis.NoHT simulations show very slow production of BrO (Fig. 7a) from HBr (Fig. 7b) with a maximum of BrO~1.2 $10^{14}$ (Fig. 7a). This is consistent with previous modelling studies (e.g. Roberts et al., 2009) showing the crucial role of species formed at vent to give a kick-start to the bromine cycle. In D.Emis.NoHT simulation, the bromine cycle is not complete before night and HBr, $O_3$ (Fig. 7c) and NOx (Fig. 7d) depletion is weak.

D.Emis.NoSulf and D.Emis.Sulf2 are used to analyse the importance of primary sulphate. Without sulphate in the emissions (D.Emis.NoSulf), the bromine cycle is more efficient than in D.Emis.NoHT but is still much slower than in D.Ref. BrO is still increasing just before night with a maximum of only ~4.2 $10^{14}$ indicating that BrO net production is likely not completed during daytime. In D.Emis.NoSulf, the sulphate aerosols required for the heterogeneous reactions are only formed from volcanic $SO_2$ emissions through its reaction with OH (secondary sulphate). This process takes time and thus slows down BrO

increase. Still this shows that these secondary sulphate aerosols from $SO_2$ emissions play a significant role when the plume is ageing. When primary sulphate emissions are doubled (D.Emis.Sulf2), there is a slight increase of BrO maximum reached about 0.5 hour earlier compared to D.Ref. Higher primary sulphate concentrations enhance heterogeneous reactions and thereby speed up the production of BrO from HBr (Fig. 7b) and the depletion of ozone (Fig. 7c) and NOx (Fig. 7d). However, the differences between D.Ref and D.Emis.Sulf2 are not very large, showing that the primary sulphate/$SO_2$ ratio chosen in

D.Ref is sufficient to provide a rapid bromine explosion. Note that the sulphate from background air was not accounted for in the initial conditions of our simulations in order to only analyse the effect of the volcanic emissions. If background sulphate is used in the simulations, it adds to the primary sulphate and therefore can increase BrO net production from heterogeneous reactions. In this study aerosol is dominated by the primary sulphate emissions, but background and secondary sulfate likely play an important role in the halogen chemistry of dispersed volcanic plumes and should be considered in future 3D regional

and global simulations.

D.Emis.NoBr and D.Emis.Br50 simulations test the sensitivity to the emission of Br radicals produced from high temperature processes. If no Br emission is assumed (D.Emis.NoBr), the production of BrO follows a curve similar to D.Ref but with a shift of ~2.75 hours later (Fig. 7a). When the partitioning of Br/HBr in the emission is assumed to be 50/50 (D.Emis.Br50) instead of 25/75 (D.Ref), the BrO/$SO_2$ maximum is slightly higher and occurs just at the end of the eruption, 1.25 hour earlier

than in D.Ref. Half of HBr is already in Br form in D.Emis.Br50 simulation, and BrO the production rate is as expected increased in this simulation. Note that this maximum is as high as in the D.Emis.Sulf2 simulation, showing that both primary

sulphate and Br emissions can be as important to rapid BrO formation. However, because Br concentration has a direct effect on BrO production while sulphate has an indirect effect through heterogeneous reactions, the maximum in D.Emis.Br50 is reached about one hour earlier than in D.Emis.Sulf2. The time evolution of the concentrations of HBr and ozone in D.Emis.NoBr and D.Emis.Br50 is well correlated with the efficiency of production of BrO, similarly to other sensitivity tests. Surl et al. (2021) tested in their 3D simulations at 1 km resolution the impact of Br and other radical emissions, also showing that an emission with no radical emissions leads to a delayed formation of BrO.

The last sensitivity test for emissions is the D.Emis.NO simulation in which NO (nitric oxide) emissions are added. For the first hour of the eruption, BrO formation is slower in D.Emis.NO than in D.Ref. This is consistent with previous modelling results (Roberts et al. 2009, Jourdain et al. 2016, Surl et al. 2021). But later, there is a more rapid BrO formation with a maximum value close to that of D.Ref, but this is reached just at the end of the eruption (Fig. 7a). This is the only simulation in this sensitivity suite that shows a full consumption of HBr (Fig. 7b) at the end of the eruption. The D.Emis.NO bromine partitioning in Fig. 8, shows that during the emission, $BrONO_2$ concentration is higher compared to D.Ref due to the additional NOx leading to enhanced HOBr (formed from $BrONO_2$ hydrolysis). This speeds up the depletion of HBr (Reaction R12). Jourdain et al. (2016) found the same behaviour in their 3D regional simulations. The $BrO/SO_2$ ratio reaches its highest values for D.Emis.NO, but this is not due to increased BrO mixing ratios that are not the highest values across all simulations. This is because in this test, adding NOx to the emissions favours reaction (R12) to (R11) leaving more OH (Fig. 7e) to react with $SO_2$ and thus leading to lower $SO_2$ concentrations compared to D.Ref.

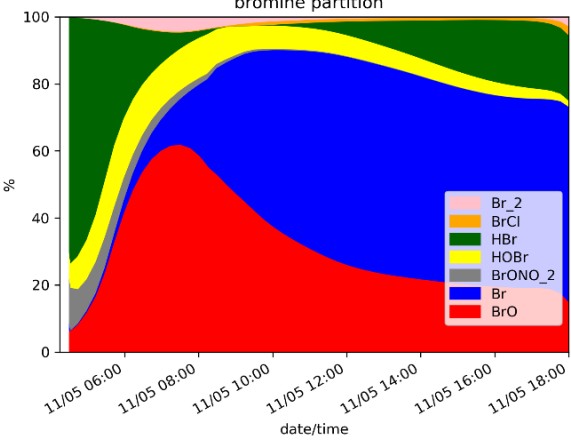

**Figure 8: Similar to Fig. 3 but the simulation D.Emis.NO.**

In summary, except for D.Emis.NO, the sensitivity tests on the emission composition find that $BrO/SO_2$ ratios (Fig. 7f) are correlated with the time variations of BrO concentrations (Fig. 7a). Apart from D.Emis.NoHT and D.Emis.NoSulf, $BrO/SO_2$ ratios reach values from $1.81 \times 10^{-4}$ to $2.02 \times 10^{-4}$, corresponding to a realistic range of values as compared to the compilation of observational data for Mt Etna (Gutmann et al., 2018).





The efficiency of the bromine cycle is largely dependent on the input emissions in MOCAGE-1D, a finding that is consistent with previous studies. Here we show a particularly important role of primary sulphate and demonstrate that the impact of changes in the emission composition that can be larger than those provided by the use of the plume parameterization.

### 680  5.3.3 Sensitivity to the effective radius

Two simulations test the sensitivity to the choice of the effective radius of sulphate aerosols: D.Reff.0.7 and D.Reff.1.0 with $R_{eff}$= 0.7 µm and 1.0 µm, respectively, instead of 0.3 µm in D.Ref. The time evolution of BrO partial column concentrations and BrO/SO$_2$ are shown in Fig. 9. Increasing $R_{eff}$ gives lower BrO and BrO/SO$_2$ maximum occurring later. $R_{eff}$ is used to define the total surface of aerosols, which is one of the parameters of the heterogeneous reaction rates (reactions (R5a), (R5b) and

(R6)). For a defined sulphuric acid concentration, assuming a larger effective radius leads to a smaller total aerosol surface and therefore to lower heterogeneous reaction rates (see Supplement material for more detail). This explains that BrO net production is slower as $R_{eff}$ increases, leading to less HBr, ozone and NOx depletion (not shown). This is consistent with results of Fig. 7a for the D.Emis.Sulf2 showing earlier and higher BrO and BrO/SO$_2$ maxima with respect to D.Ref. In D.Emis.Sulf2 we assume twice as much primary sulphate concentrations compared to D.Ref, leading to an increase of the total surface of

aerosols and thus to a more efficient production of BrO *via* higher rate constants of the heterogeneous reactions.

Compared to the plume parameterization simulations (section 5.2.2), the time evolution of BrO and BrO/SO$_2$ in D.Reff.0.7 is close to D.Plume.0.1, whereas D.Reff.1.0 gives a lower maximum in BrO. This shows that the choice of $R_{eff}$ can be even more important than the use of the plume parameterization.

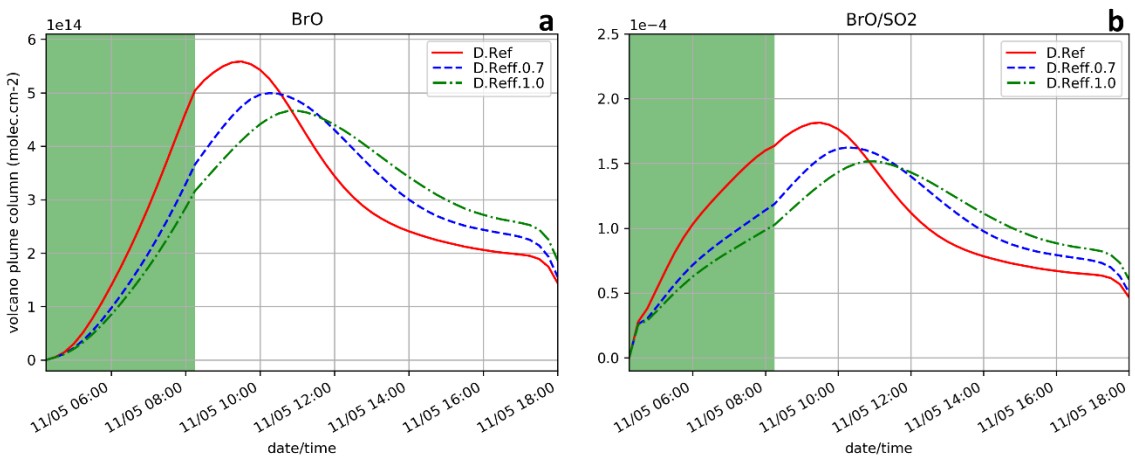

**Figure 9: Similar to Fig. 4a and 4f but the simulations testing the sensitivity to the effective radius of sulphate aerosols: D.Ref, D.Reff.0.7 and D.Reff.1.0.**



### 5.3.4 Sensitivity to eruption height

D.Alt.9.5 and D.Alt.7.5 simulations test the sensitivity of the results to the top altitude of the eruptive emissions, 9500m and 7500m respectively, instead of 8500m for D.Ref. Note that for these tests, the figure for $BrO/SO_2$ is not provided since the number of model levels used to calculate the partial columns on the emission levels vary and thus the column of $SO_2$ is not comparable between the experiments because of the background profile of $SO_2$.

Figure 10 shows that when the top altitude increases, the maximum of BrO occurs a bit later (one hour) and with slightly lower 705  values ($0.3 \ 10^{14}$ molecules.cm$^2$ difference). This is because at the model levels where most of the emissions are set (top third part of the profile), the concentrations of oxidants are lower at higher altitudes, leading to a lower BrO production. Here, the vertical variation of the concentrations of oxidants is the main driver of the changes of the bromine cycle efficiency. Thus, injection altitude is shown to be an important parameter.

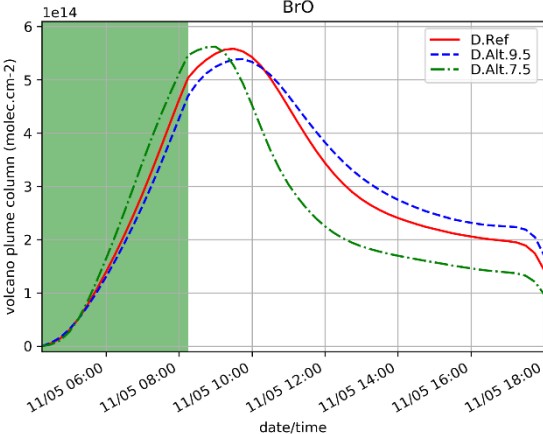

**Figure 10: Similar to Fig. 4a but the simulations testing the sensitivity to the height of the emissions: D.Ref, D.Alt.9.5 and D.Alt.7.5. Here, the quantities are integrated vertically on the emission levels: 3300m-8500m for D.Ref, 3330m-9500m for D.Alt.9.5 and 3300m-7500m for D.Alt.7.5.**

### 6 Conclusion

The formation of BrO in volcanic plumes is important in the budget, atmospheric fate and impacts of volcanic bromine emissions. From the volcanic emissions of HBr, BrO can be formed in volcanic plumes. Compared to HBr, BrO is not soluble and therefore can remain longer in the atmosphere as the plume disperses. This means that it can have the potential to undergo long-range transport and have impacts further afield as shown by the regional modelling study of Jourdain et al. (2016), the only study at the regional scale yet published. The present paper has the general objective to prepare the implementation of 720  volcanic halogen chemistry in the 3-D chemistry-transport model MOCAGE for regional and global simulations. For this, the





1D version of MOCAGE is used to study the time evolution up to 20 hours of a volcanic eruption containing halogen compounds. The 1-D framework allows us to make a large series of sensitivity tests on different parameters.

The 1D simulations were run with resolution of 0.5° longitude x 0.5° latitude (an intermediate resolution for regional and global MOCAGE applications). The MOCAGE chemical scheme was modified to account for the halogen cycle in volcanic plumes based on recent modelling studies (mainly Surl et al. 2021). The case study is the 4-hour eruption of Mt Etna that occurred on 10 May 2008. The plume of this eruption was sampled with the GOME-2 spaceborne instrument on 11 May morning (Hörmann et al., 2013).

The analysis of N.Ref (reference) simulation shows results that are within the range of the BrO/SO₂ ratio observed from GOME-2, ~18 hours after the end of the eruption. The results (and sensitivity studies, outlined below) are in general agreement with previous model and observation studies of volcanic plume halogen chemistry. The halogen chemistry developed in MOCAGE-1D and based on previous studies, is able to produce a realistic bromine partition during both daytime and night, and the following morning when BrO was observed (Hörmann et al., 2013). Additionally, to account for the fact that volcanic plumes close to the vent have a size much smaller than the typical MOCAGE global/regional model resolutions (from 2° to 0.1°), we tested a simple sub-grid scale parameterization based on Grellier et al. (2014) to represent the chemistry in the plume within the model box. For this, a Plume box is defined as a small box within the Model box (Plume box volume = 1/400 Model box volume) in which all the emissions are injected. We upgraded this parameterization to make it more realistic by implementing mixing between the Plume box and the Model box continuously during and after the eruption. The results show that the use of this plume parameterization slightly slows down BrO formation and its maximum concentration when the dilution rate decreases but still remains within the GOME-2 estimates of BrO/SO₂ ratio. The results of the plume parameterization reflect the important control of oxidants on BrO formation (that in turn depletes atmospheric oxidants). They are consistent with previously reported observed and modelled decrease of BrO/SO₂ in the core of volcanic plumes where there are less oxidants than at the edge of the plume (Bobrowski et al. 2007, Roberts et al., 2018). Because night comes just after the end of the eruption and stops BrO production before being complete, we also run simulations starting the eruption at the beginning of the day on 11 May at 04:15 UTC (D. simulations). Findings from the daytime results without and with the plume parameterization are fully consistent with the simulations including night (N. simulations), in terms of partitioning between halogen species e.g. BrO, Br, HOBr, and the role of oxidants.

Apart from the issue of spatial resolution, there are other sources of uncertainties in the modelling of halogen-rich volcanic plumes. Previous studies showed that the quantity and composition of the emissions used as input in the model are important. We first tested a lower total bromine/SO₂ emission ratio (resp. a lower emission flux for all species) giving a more (resp. a less) efficient BrO production. These results are consistent with the literature. Note that for these two sensitivity simulations, the impact of applying the subgrid scale parameterization is more important than in the reference simulation and leads to an increase of BrO/total bromine ratio in the first hour of simulation. Secondly, we tested the sensitivity of the model to the emission, including species formed near the vent at very high temperature when magmatic air first mixes with atmospheric air. We show from sensitivity simulations on emitted Br and primary sulphate that both are important for a rapid BrO production,





but primary sulphate aerosols are more important because they are needed for the heterogeneous reactions which are dominant in the bromine explosion. We also run a test adding NO emissions, as assumed in several previous studies. In this case, the $BrO/SO_2$ increase is slower compared to the reference in the first hour but then gives a more efficient net production of BrO (higher $BrO/SO_2$ ratio).

Sensitivity tests on the choice of the effective radius for the sulphate aerosols and of the top altitude of the plume highlight

that these parameters are important for the bromine cycle because of the role of aerosols in the heterogeneous chemistry and of the vertical variability of oxidant concentrations available for the bromine cycle, respectively. Compared to the sub-grid scale plume parameterization on the bromine cycle, the impacts of the other sensitivity tests are at least comparable and sometimes more important. Knowing that there are large uncertainties on emission composition, $R_{eff}$ and sometimes on the plume altitude, we find that the subgrid-scale parameterization is not the model setting that will be most important in

MOCAGE-3D global/regional simulations. This parameterization tends to slightly delay and weaken BrO net production for the emissions of the reference simulation, with similar results for different subgrid-scale parameterization settings. It is possible to achieve a similar behaviour with a larger $R_{eff}$, and/or lower primary sulphate concentrations and/or lower Br/HBr ratio in the emission (see summary Tables 6 and 7). In the case of a lower emission flux and of lower total bromine/$SO_2$ ratio, the parameterization tends to increase the BrO/total bromine ratio in the first hour of simulation. A similar effect could be

simulated by using higher primary sulphate concentrations and/or Br/HBr emission ratio. We chose here to propose a 1D parameterization simple enough to be possibly implemented in 3D at a fairly low cost. Whilst a more advanced 3D plume-in-grid approach could more fully represent the subgrid scale effects, our sensitivity tests show that other sources of uncertainty e.g. in the emission composition are equally or more important.

This study also gives insights for MOCAGE-3D improvements in view of regional and/or global studies. OH emissions from

high temperature processes were not included because it is a diagnostic species in MOCAGE. An upgrade of the chemistry in MOCAGE-3D is on-going and will allow to include these emissions that would increase the BrO formation efficiency in the early stages of the plume. The MOCAGE-1D study assumed a unique value for the effective radius of the sulphate aerosols. In reality, this effective radius is likely to evolve as the plume ages. This is why we plan to use for the 3D simulations the version of MOCAGE including the secondary inorganic aerosol module coupled to the chemistry scheme (Guth et al. 2016).

The model main timestep (15 minutes), is not short enough to represent the chemical evolution in the very early stage of the plume chemistry, thus the model is not suited to perfectly simulate $BrO/SO_2$ ratio during the first 30 minutes/1 hour of emissions as has been done using 0D/1D or local scale simulations (e.g. Roberts et al. 2018 and references therein). Given that the final aim is to run MOCAGE-3D simulations to analyse the regional/global impact of volcanic halogen emissions and not to analyse their fate at fine temporal and spatial resolutions, this is not considered as a problem. Overall, the MOCAGE-1D

simulations are consistent with previous studies showing that the version of the MOCAGE chemistry developed for this study is suitable to study halogen chemistry in volcanic plumes and brings a first step towards global modelling of the impacts from volcanic halogen emissions to the troposphere.



**Table 6: Summary of the influence of the plume parameterization on BrO.**

| Simulation | Impact of the plume parameterisation for X=0.1 | | | |
|---|---|---|---|---|
| | Behaviour of BrO concentrations in the first timesteps | Increase of BrO after the first timesteps | Max of BrO concentration | Time of max BrO concentration |
| Reference emisions | Slight increase | Weaker | Decrease | Later |
| Low HBr/SO$_2$ emission ratio | Increase | Weaker | Similar | Similar |
| Low emission flux | Increase | Weaker | Similar | Similar |


**Table 7: Summary of the influence of the emission composition and plume altitude on BrO.**

| Sensitivity simulation | Change with respect to the reference simulation | |
|---|---|---|
| | Max concentration of BrO | Time of max BrO concentration |
| ↑ mass of primary sulfate aerosol emissions | Increase | Slightly earlier |
| ↓ mass of primary sulfate aerosol emissions | Strong decrease | Much later |
| ↓ surface area of primary sulfate aerosol | Decrease | Later |
| ↑ Br/Total bromine emission ratio | Slight increase | Earlier |
| ↓ Br/Total bromine emission ratio | Similar | Later |
| Addition of emissions of NO | Similar | Earlier |
| ↑ altitude of the plume | Slight decrease | Similar |
| ↓ altitude of the plume | Similar | Slightly earlier |

**Code availability**

This paper is based on source code that is presently incorporated in the MOCAGE-1D model. The MOCAGE-1D source code, which derives from MOCAGE-3D source code, is the property of Météo-France and CERFACS and is not publicly available.

This is because MOCAGE-3D includes routines protected by intellectual property rights. Note that Météo-France plans to make available the part of MOCAGE-3D and MOCAGE-1D models dealing with gaseous chemistry as a free software in the future. However, the newly developed/modified code in the MOCAGE-1D version is made available to the editors and the referees for the review process (https://zenodo.org/record/6876348, doi: 10.5281/zenodo.6876348). This code will be available in the 2023 release of MOCAGE-1D.

**Data availability**


All data corresponding to the results presented in the paper can be downloaded from https://mycore.core-cloud.net/index.php/s/4jbUu6aHG4yZQN0.



**Supplement link**

**Author contribution**

VM, BJ, LG and RPV conceived the general methodology. RPV, HN, BJ, TJR, LS, PDH and JG contributed to build the 1D-MOCAGE model version and the visualisation tools. AA provided emission estimates. BJ provided the chemical initial conditions. VM, TJR and PDH designed the simulation settings. VM ran the simulations. VM prepared the paper with contributions of all co-authors.

**Competing interest**

The authors declare that they have no conflict of interest.

**Financial support**

We are grateful for support from the ANR Projet de Recherche Collaborative VOLC-HAL-CLIM (Volcanic Halogens: from Deep Earth to Atmospheric Impacts), ANR-18-CE01-0018.

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
