# Peer review of "Halogen chemistry in volcanic plumes: a 1D framework based on MOCAGE-1D (version R1.18.1) preparing 3D global chemistry modelling"

_Geoscientific Model Development, 2022_

## Author Comment (AC1)

Dear Dr Ham,

Thank you for your recommendations.

Regarding the code, only the newly developed code was provided because we had understood that it was possible to do it. But we fully understand the need for the reviewers to access the full code to check if it is reproducible. The full code is now available on Zenodo (doi: 10.5281/zenodo.7298580, https://zenodo.org/record/7298580) with instructions to compile and run it, and the input files required to run the simulations discussed in the paper. The full code has restricted access to Dr Astrid Kerweg (the editor of the paper), to the reviewers and to you.

Regarding the data archive, following your suggestion, the output files from the simulations are now available on Zenodo (doi:10.5281/zenodo.7299080, https://zenodo.org/record/7299080) instead of the mycore database which was previously used. For the Plume type of experiments, two set of files are produced by MOCAGE R1.18.1. The files in the Zenodo database need to be combined to get the concentrations which are analysed in the paper. In the previous set of files made available on mycore database, these files were already combined. The method for recombining the files in the Zenodo database is explained in the README.txt (available with the code) and also in the data archive information.

The last point is that in the course of preparing the code, a small error was found. This error does not affect most of the simulation results at all while for a few simulations there are only small changes which do not affect the analysis and conclusions of the paper. We enclose here the only figure where it is possible to see these changes. In Figure 2, it can be seen for N.Plume.0.1 that HBr is a bit lower in the new figure than in the original one and that there is a slight difference in BrO and BrO/$SO_2$ at the end of the simulation time. We will inform the reviewers in our answers and will update the figures in the next manuscript version to be produced. Note that the Zenodo (10.5281/zenodo.7299080, https://zenodo.org/record/7299080) archive of data files contains the new version of the data.

Best regards,

V. Marécal and co-authors

[Figure]

Figure 2 (new version)

---

## Author Comment (AC2)

**Response to reviewer 1** (Citation: https://doi.org/10.5194/gmd-2022-180-RC1)

We thank you for your useful comments that helped us improving the paper. Our response is organised as follows. After each of your comments (bold) you will find the authors' response followed, if needed, by the changes that were made in the manuscript (in blue). In the revised version of the manuscript, the changes that are significant are coloured in blue to help identifying new contents. The paragraphs that have been moved to a different place in the manuscript are coloured in green.

Note that in the course of preparing the code in response to the editor's comment, a small error was found. As explained to the editor, this error does not affect most of the simulation results at all while for a few simulations there are only small changes which do not affect the analysis and conclusions of the paper. The only figure where it is possible to see these changes is Figure 2 for N.Plume.0.1: HBr is a bit lower in the new figure than in the original one and there is a slight difference in BrO and BrO/$SO_2$ at the end of the simulation time. The revised manuscript includes the new (error-free) figures.

Note also that following reviewer 2's recommendation, we have given less emphasis on the comparison of the simulations with the GOME-2 observations.

**First, my abject apologies for letting my review slide past the deadline. The authors deserved a more timely review than I provided. I began several weeks ago, reading from the beginning, and making typical, small-correction notes for the authors to improve the manuscript's readability. Then at L326, I "hit the wall" in terms of what the model was doing. I did not know how to continue with the review when I was unsure as to the physical representation of this new model.**

We understand your concerns about what is called the "plume sub-grid scale parameterization" in the paper and we answer your remarks and questions below. Before, we want to stress that the first and main aim of the paper is to evaluate if the halogen chemistry developed in MOCAGE, that is based on previous studies, is able to produce a realistic bromine partitioning at a typical MOCAGE-3D horizontal resolution (0.5° latitude x 0.5° longitude). The use of the MOCAGE-1D configuration, with 0D boxes stacked on top of each other following the model vertical resolution, is not meant to be a fully realistic configuration but to give us a framework that allows an easy interpretation of the chemistry results and the possibility to run a large number of sensitivity simulations at low computing cost that would not be possible to run with the MOCAGE-3D version. Most of the sensitivity simulations discussed in the paper are part of the evaluation of the volcanic halogen chemistry scheme since their results are compared to similar sensitivity tests published in previous modelling studies. The results discussed in the paper show that the chemistry scheme implemented in MOCAGE for volcanic halogen chemistry is realistic and consistent with the literature.

The second and secondary aim of the paper is to try to address the possible effect on the bromine explosion cycle of the assumption that chemical species are homogeneously distributed within each model grid box while we know that the typical size of a volcanic plume during its early stage is much smaller than the MOCAGE horizontal resolution. This is why we propose a simple method that we called a "plume parameterization" to account for the fact that the chemistry processing of the volcanic emissions takes place in a smaller volume than the model grid box. Your remarks made us realise that we did not present this part of the work in the right way in the original version of the paper. The objective is not to develop a parameterization that fully represents the plume evolution but to propose a simple method to test if assuming that the volcano emissions are instantaneously fully mixed within the model grid boxes affects the chemical processing of the halogen emissions. The basis is that we know from observations and modelling that the efficiency of the bromine explosion varies depending on the concentrations of oxidants available and therefore from the mixing with background air that provide these oxidants (e.g., Bobrowski et al., 2006; Jourdain et al., 2016; Roberts 2018, Surl et al. 2021).

We fully acknowledge that it was not clear all throughout the paper and in particular in the introduction section that:

1. the main aim of the paper was to evaluate if the halogen chemistry developed in MOCAGE is able to produce a realistic bromine partitioning at a typical MOCAGE-3D model box size and

2. the secondary aim was to address the 'plume effect' (i.e., locally concentrated volcanic emissions) on the chemistry processing and in particular on the bromine partitioning.

We have revised the introduction, section 4.2 and the conclusion of the manuscript to make clear the points explained above.

Finally, we would like you to note that the results of the paper indicate that the impacts on the bromine cycle of the composition of the emissions are often comparable and sometimes more important than the effect of assuming that the chemical processing of the emissions occurs in a smaller volume (plume volume) than the model grid box. Knowing that there are large uncertainties on the emission composition, on the typical size of volcanic sulphate aerosols and sometimes on the plume altitude, we find that what we called "the subgrid-scale parameterization" is not the model setting that will be most important in future MOCAGE-3D global/regional simulations.

**The intervention by the Editor Ham regarding the paper's suitability pushed me to quickly finish this review so that it may still be useful. I would like to see this work published, but it does need some more clarity for the science, and documentation of the model as noted by the Editor.**

**First, this appears to be a 0-D box-in-box model since there is no interaction in the vertical?**

Yes you are right. This is what is explained in the model description by:

"The 1D configuration also assumes no transport horizontally and vertically (unlike the 3D version). Thus, the boxes constituting the vertical column are not interacting with each other and can be considered as an ensemble of independently piled 0D boxes."

**Is there not interaction between the overhead plume and the photolysis rates? There appears to be no other 1D connection.**

You are also right. The only connection between the overhead plume is only through photolysis rates. This is now clearly stated in the revised manuscript.

**Second, from the three bullets (L344-349) it is not clear that any chemistry is calculated within the P box. I presume it is – Yes it is so stated in Figure 1.**

Yes the chemistry is calculated within the P-box as explained in figure 1. The piece of text describing the different steps was not clear. We have revised it to make it fully consistent with figure 1.

**Third, is the plume constantly injected over multiple time steps? That would seem odd as the upper layers are moving across the volcano, not hovering. From Figure 1, the P-box stays exactly over the emission plume for the whole time of the eruption. That makes no sense given normal atmospheric winds.**

Yes, the plume is constantly injected over multiple time steps. You are right that this setting is not fully realistic because there is no actual plume transport by horizontal winds. Still our method indirectly represents the transport of the plume within the Model box by the fact that we simulate the progressive dilution of the plume with the background air of the Model-P box. The Plume box is only used to calculate the chemical processing of the emissions within an air volume typical of the size of a volcanic plume. Ultimately, we are interested to analyse the effect of this processing on the final partitioning of the bromine species at the scale of the Model-box.

This information has been included in the revised manuscript at the end of Section 4.2.

**at time T1 you have a volcanic plume (size = P-box) going into the M-box, OK**

**at time T2, you have done chemistry on both boxes and then mixed at some rate**

**in your step 3 (not T3) you state that you mix unidirectionally from P-box to M-box, does not the P-box shrink? but then in step 4, you mix M-box air back into P-box. OK**

**but at step 5 you add more fresh volcanic emissions to the P-box does that not include new air mass?**

**does the P-to-M mixing rate control the rate of new emission flux into P-box?**

At step 5 we add in the P-Box the molecules that have come from the volcanic emission: for each species emitted we calculate the number of molecules injected in the P-box from the emission flux multiplied by the 15 min timestep. Therefore, the new emission flux into P-box is only related to the emission flux that is set according to Table 2 (Table 1 in the revised version).

The legend of the figure has been updated with this information. In addition, to make the text clearer, we have completely re-written the description of the plume parameterization to be completely consistent with Figure 1 by using the same numbering for the steps and by detailing them.

**This model implies that the outflow of chemically processed P-box air keeps going into the same M-box air. Possible, but sounds difficult.**

We agree that the model implies that what is processed in the P-Box keeps into the same M-Box. In reality, the plume mixes with time with the background air leading to the progressive spread of the plume. The spread on the plume will reach at some point a horizontal size larger than the M-Box, But because the M-box is large (0.5° latitude x 0.5° longitude), this effect would only become significant after several hours up to 1-2 days. The possible impact of neglecting this effect is taken into account in the analysis of the results.

**Fourth, the mixing between P box and M-P box should be based on a mixing frequency defined by the inverse time to mix the P box (e.g., 4 /hour for 15min).**
The wording we used for designating X was wrong and led to a misunderstanding. X should not be called the dilution rate but a dilution coefficient since this is not a rate. This has been modified accordingly in the revised manuscript. In addition, we now provide in the text (section 4.2) the mixing rate per hour corresponding to the two values of X chosen (X=0.3 and X=0.1).

**Presumably you conserve air mass and mix equal parts air mass of the M-P and P boxes and then put it back into both boxes? I cannot tell if that was done. YES, after reading the Fig 1 caption, it is clear stated. You need to be more precise in the sentences describing this in the text and not rely on the Figure caption.**

As you suggested and as explained above, to make the text more in line with Figure 1 and more complete and clearer, we have changed the presentation of the plume parameterization in the text. We use in the revised manuscript the same numbering of the steps as in Figure 1 and describe each of them.

**Fifth, you should really show that your results converge as you reduce the time step. That shows you are reasonably modeling a continuous process.**

As you suggested, we have run simulations with a timestep down to 1 min instead of 15 min. The partition results for the 1 min tests are shown below and can be compared to figures 3 and 5 of the paper. The results are very consistent between the 1 min and the 15 min tests. The only difference is at the first 1.5h of the simulation where there is a BrO production a bit higher with the 1 min timestep. This is due to the fact that, at the very early stage of the plume, there are enough oxidants available to produce very rapidly BrO and with a 1 min timestep this happens even more quickly than with a 15 min timestep. This behaviour is consistent with what we expect from model results and observations already published (Bobrowski et al., 2007; Jourdain et al. 2016, Rüdiger et al. 2021): in the first hour there is at first a strong and rapid increase of BrO in the centre of the plume that then is stopped because of the bromine-explosion cycle having consumed the oxidants. At the edges of the plume where there is mixing with atmospheric air or when the plume dilutes while ageing, then the atmospheric air provides the oxidants needed to increase BrO production again if there is still HBr

available. The behaviour of the results using the plume parameterization mimics this behaviour of a strong increase then a pause and then again an increase when the mixing with the M-box air provides new oxidants to promote BrO formation. So unlike what was said in the original version of the paper (lines 509-511), the behaviour of the $BrO/SO_2$ at the beginning of the simulation with a 15 min timestep is realistic and consistent with the simulations with a 1 min timestep. The text has been revised accordingly (section 5.1 and conclusion).

[Figure]

| Similar to Fig 3 but for a 1 min timestep | Similar to Fig 5 but for a 1 min timestep |

**That is why mixing rates need to be in per hour.**

See answer above regarding the dilution rate.

**Basically the model looks sort like a smokestack model, but even there the exhaust is constantly encountering new background (M-box) air. Putting emissions of trace gases into the P-box is maybe convenient, but in reality these emissions come with an air mass that must be incorporated into the P-box. Is this the way you would model a standard smokestack plume?**

The emissions that we use as input in the model are the result of the chemical processes at high temperatures (> 500°C) that occur when the magmatic air first mixes with the atmospheric air at vent (called "effective source" in previous studies). This means that we take into account in the emissions the atmospheric air that is first mixing with the magmatic air. Then, this is the choice of the experiment setup (with and without the plume parameterization) that drives further mixing of the emissions from this effective source with the atmospheric air contained in the M-box.

**So, overall, I am not sure what kind of plume you are modeling. The physics of injection and mixing seem not to be realistic for volcanic plumes. I am readily willing to be convinced otherwise if the authors can make a clear case.**

As explained above, we agree that we did not present well enough what we call the Plume parameterization in the original version of the paper. The aim is to propose a simple method to test if assuming that the volcano emissions are instantaneously homogeneously mixed within the model grid boxes affects the chemical processing of the halogen emissions and in particular the bromine explosion cycle. We have revised the whole manuscript accordingly.

**Comments below may be helpful for language use when revising this paper.**

**Title: awkward, try maybe: Halogen chemistry in volcanic plumes: a 1D framework based on MOCAGE-1D (version R1.18.1) preparing 3D global chemistry modelling**

We do not understand because this is already the title of the paper.

**L20 – maybe "a 1-D single-column configuration…."**

Done

**L28 – how about also to the background atmospheric conditions?**

The background conditions have an influence on the plume chemistry. This is illustrated by the simulations done at different times of the day (L538-542 in the GMDD version but not mentioned in the abstract) and by the sensitivity simulations on the altitude of the plume. We have added this information in the abstract of the revised version.

**Abstract – Overall, it is too long, can you shorten?**

The abstract has been shortened as suggested.

**L42 – Iodine clearly eats O3, does CL do much in the troposphere? (not sure)**

Chlorine is much less efficient than bromine and iodine for ozone depletion. Sherwen et al (2016) estimated that the tropospheric ozone consumption by halogens is 57% from iodine, 39% from bromine and 4% by chlorine at the global scale.

The sentence was changed to "Bromine, and to a much lesser extent chlorine…."

Sherwen, T., Schmidt, J. A., Evans, M. J., Carpenter, L. J., Großmann, K., Volkamer, R., Saiz-lopez, A., Prados-roman, C., Mahajan, A. S. and Ordóñez, C.: Global impacts of tropospheric halogens ( Cl , Br , I ) on oxidants and composition in GEOS-Chem, , 12239–12271, doi:10.5194/acp-16-12239-2016, 2016.

**L104 – 'no possibility' is too strong, how about 'no direct way to '**

Done

**L122 – I am confused, a 1D single-column model really has not horizontal resolution, yet here you talk about it grid size and a sub-grid parameterization.**

This sentence has been changed.

**L123 – 'also' not needed.**

Done

**L135 – Make its shorter and easier to see the 3 things:**

**There are three reasons behind the choice of this volcanic eruption: (1) Mount Etna is one of the largest known emission sources of halogens (Aiuppa et al., 2005); (2) the Etna volcano is also continuously and extensively monitored by INGV (Istituto Nazionale di Geofisica e Vulcanologia) including gas composition needed for the model; and (3) satellite observations above the Mediterranean region are available.**

From your suggestion and the remarks of Reviewer 2 we have changed the text to:

There are two reasons behind the choice of this volcanic eruption: (1) Mount Etna is one of the largest known emission sources of halogens (Aiuppa et al., 2005) and (2) the Mount Etna volcano is also continuously and extensively monitored by INGV (Istituto Nazionale di Geofisica e Vulcanologia) including emission flux estimation and gas composition needed for the model. In addition, satellite estimations of BrO and $SO_2$ of the plume are available on 11 May and have been used in addition to the literature to evaluate if MOCAGE 1D simulations give plausible values.

**L146 – 'passive' = 'non-eruptive'**

Done

**L150 – "The composition of Mount Etna plumes has extensively been characterised before this case study by both in situ (e.g.,….**

Done

**L152 – drop the "as"**

Done

**L153- 'location' instead of 'space' ??**

Done

**L155 – ' … more distal, safe locations, are…'**

This sentence has been removed in the revised manuscript (from reviewer 2's comments).

**L158 – 'such' instead of 'similar'**

This sentence has been removed in the revised manuscript (from reviewer 2's comments).

**L161 – eruptions are available, and none for 10 May**

This sentence has been removed in the revised manuscript (from reviewer 2's comments).

**L169 – I would drop the clause: 'being representative of Etna emissions'**

Done

**L176 - 'al., 2021). MOCAGE is developed"**

Done

**L177 – drop ' Due to the low computational cost', just say simply that: "This 1-D configuration of MOCAGE allows us to make a large set of sensitivity tests on the…"**

Done

**L179 – very confusing ("It does not…"), do you mean: "MOCAGE 1D does not focus on the very early stages…. " then stop and delete "but to …." this is already said or implied.**

Done

**L182 – "corresponds here to the vertical column" I do not agree with this. You need to make it clear that the 1-D column model does not connect across vertical layer with transport or photolysis (? does it). It starts as a 1-D column, but the layers shear out an separate as they would in 3D. Right? OK OK, I see this in L187, so maybe combine this information.**

Done

**L190 – the problem is the entrainment mixing on the way up. You should consider doing LES models or other models to find the entrainment factor as a function of altitude**

As explained in the answers to your major comments, with our simple plume parameterization, we do not intend to model the detailed evolution of the plume entrainment mixing but to evaluate if there are major changes in the bromine processing when the emissions are concentrated in the plume or if they are assumed to be homogeneously distributed in the model gridbox.

**L195 – do not keep mentioning 'future' – "The 1-D configuration of MOCAGE is designed so that the chemistry model developed for volcanic emissions can be seamlessly inserted into MOCAGE-3D.**

Done

**L201 – " MOCAGE-1D start with those in MOCAGE-3D,**

Done

**L233 – I am looking forward to a plot of chemistry vs time showing this Br explosion**

We understand from this remark why you asked us to add a plot. This is because the bromine explosion was not explained well enough. The bromine explosion corresponds to the very rapid conversion of HBr that is emitted from the volcano into reactive species, BrO in the first place. Several observation and model studies have shown this very rapid increase of BrO that can occur within a few minutes depending on the plume emission intensity and composition, on the dilution of the emissions and the composition of the background air (see for instance Roberts et al. 2009, 2014). Instead of including a plot showing the bromine explosion which would require to use a plot from another paper, we have added some text to better explain the bromine explosion in the paragraph after reaction (R12).

**L245-256 – All this makes sense and seems logical.**
**L255 – 'of volcanically derived sulphate' The volcano emits sulphate or SO2? if both (below) then make this clear**

The volcano emits both $SO_2$ and sulphate. This is made clear in the revised manuscript.

**L262 – "The RELATIVE TRACE GAS composition in the plume is …Table 1.**
 **[What is unclear here is the absolute concentration in the plume at each level?]**

This sentence was not clear and has been changed.

The molar ratio of the main magmatic gas species emitted by Mount Etna volcano on 14 May 2008 is given in Table 1.

Regarding the vertical distribution of the emissions, the information was given at lines 198-200. We have revised the sentence which was not fully clear.

For eruptions, the emissions are spread from the volcano crater altitude to the top height of the plume following an "umbrella" profile as in Lamotte et al. (2021), with an injection of 75% of the emissions in the top third of the plume. This represents the fact that most of the mass emitted during an eruption is in the top part of the plume.

**L266 – the high temperature (magma?) chemistry should be already taken into account in Table 1.**
**L276 – 'given in Table 2' – this really needs to be combined with Table 1.**

As suggested we have merged the two tables in one and revised the description of the table to make it clearer the difference between the molar ratios which come from the 'raw' magmatic air and those which are input into the model and that come from the processing at vent at high temperature of the 'raw' magmatic air when it first mixes with atmospheric air in the very first moment of the emission.

| Species | Molar ratio to $SO_2$ of the magmatic gas composition from Mount Etna volcano on 14 May 2008 | Molar ratio to $SO_2$ used as input in the model and resulting from the processing at high temperature at vent | Eruption emissions in tons between 14.15 and 18.15 UTC used as input in the model |
|---|---|---|---|
| $SO_2$ | 1 | 1 | $8.00 \ 10^3$ |
| HCl | 0.3 | 0.3 | $1.37 \ 10^3$ |
| $H_2S$ | $6.6 \ 10^{-3}$ | $6.6 \ 10^{-3}$ | 27.0 |
| CO | $3.1 \ 10^{-3}$ | $3.1 \ 10^{-3}$ | 10.9 |
| HBr | $3.28 \ 10^{-4}$ | $2.46 \ 10^{-4}$ | 2.50 |
| Br | 0 | $0.82 \ 10^{-4}$ | $8.21 \ 10^{-1}$ |
| Primary sulphate aerosols | 0 | 0.02 | $2.40 \ 10^2$ |
| $H_2O$ | 129 | | |
| $CO_2$ | 11 | | |
| $H_2$ | 0.23 | | |
| HF | 0.13 | | |
| HI | $7.7 \ 10^{-6}$ | | |

**L302 "THAT PARTICULAR eruption…"**

Done

**L307 – the aerosol mass, diam and surface area should be in Table 1+2. Do not scatter critical input parameters.**

The aerosol mass emitted is now in the new Table 1 (merged Tables 1 and 2). We do not find it useful at this stage to include within the table the information of the effective radius ($R_{eff}$) but we have added it in the table caption.

**L321 – (resp. 'D.BGD)' (spelling)**

Done

---

## Author Comment (AC3)

**Response to Reviewer 2** (https://doi.org/10.5194/gmd-2022-180-RC2)

We thank you for your useful comments that helped us improving the paper. Our response is organised as follows. After each of your comment (bold) you will find the authors' response followed, if needed, by the changes that were made in the manuscript (in blue). In the revised version of the manuscript, the changes that are significant are coloured in blue to help identifying new contents. The paragraphs that have been moved to a different place in the manuscript are coloured in green.

Note that in the course of preparing the code in response to the editor's comment, a small error was found. As explained to the editor, this error does not affect most of the simulation results at all while for a few simulations there are only small changes which do not affect the analysis and conclusions of the paper. The only figure where it is possible to see these changes is Figure 2 for N.Plume.0.1: HBr is a bit lower in the new figure than in the original one and that there is a slight difference in BrO and $BrO/SO_2$ at the end of the simulation time. The revised manuscript includes the new (error-free) figures.

Note also that following reviewer 1's recommendation, we have shortened the abstract and merged Tables 1 and 2.

**Marecal et al. evaluate volcanic plumes in the 1D-version of the model MOCAGE. They also test a sub-grid scale parameterization. The sensitivity studies are very interesting, and the chemistry scheme is adequate to describe bromine explosions.**

**General comments:**

**My main criticism is that I do not find the comparison to the Etna eruption of 10 May 2008 very convincing because there seem to be hardly any useful observations for this comparison:**

We understand your concern about the availability of observations. We want to stress that the philosophy of the paper is to make a plausible case study to test the volcanic chemistry scheme implemented in the 1D model and not a detailed analysis of the eruption. To make sure to run the model with plausible conditions, we picked this particular Etna eruption because its $SO_2$ emissions have been estimated in previous work (flux and top height of injection) and we have information from observations on the magmatic gas composition for halogens (molar ratios of bromine and chlorine versus $SO_2$). To make this clearer, we have added at the beginning of section 2:

The philosophy of the paper is to make a plausible case study to test the volcanic chemistry scheme implemented in the 1D model and not a detailed analysis of the eruption. To try to run the model with realistic conditions, we picked the particular Etna eruption of 10 May 2008 because its $SO_2$ emission flux and height have been estimated in a previous study and we have information from observations on the magmatic gas composition for halogens.

**- I didn't see any observational BrO data mentioned with one exception: a single data point from GOME-2 (2.3E14).**

Observations of BrO in volcanic plumes are fairly scarce. There are ground-based remote sensing measurements sampling plumes available from field campaigns in very young plumes close to the vent (a few hundreds meters to a few kilometres distant maximum) but they are not representative of the model resolution. There have also been in the past a few remote sensing measurements from aircraft of volcanic plumes but they are also at fine resolution and close to the vent. On the contrary, satellite derived measurements have the advantage of observing aged volcanic plumes further from the vent and at horizontal resolutions similar to the model. This is why we picked a case study for which satellite observations of BrO and $SO_2$ were available (Hörmann et al. 2013). However, since we use a 1D configuration and because satellite columns of BrO have significant uncertainties, this is not possible to do a full quantitative evaluation of the model. Still, the satellite observations are used in addition to the literature to assess if the model values are at least plausible. From your remark, we have modified the

text to put less emphasis on comparison with GOME-2 observations in the revised manuscript (removed from the abstract, section 4.1 and conclusion, comparison shortened in section 5.1).

**- Bromine is systematically below the detection threshold of FTIR (page 5, lines 158-159).**

There is no available report in the literature of HBr remote sensing by FTIR. HBr absorbs in the IR, but at spectral wavelengths at which absorption by major gas compounds dominates. HBr can therefore be only detected by in-situ sampling in plumes with alkaline traps or via direct fumarole sampling. Both techniques are not viable strategies in eruptive plumes. This motivates the use of measurements taken in the passive plume just a few days after the explosion. In the revised manuscript we do not mention anymore the FTIR technique since it does not give information relevant to our work which is focused on bromine.

**- No reports of near-downwind volcanic BrO are available for 10 May 2008 (lines 159-162).**

To determine the bromine emissions to use as an input in the model, there are two possibilities. One is to use the total columns of $SO_2$ and BrO from DOAS measurements across transects within the plume close to the vent to estimate the $SO_2$ emission fluxes and the $BrO/SO_2$ ratios (Gutmann et al. 2018, Dinger et al. 2021). However, the $BrO/SO_2$ ratio cannot be directly related to the total bromine emitted by the magmatic gas because the partitioning between the bromine species varies close to the vent due to very high temperature processes and the bromine explosion occurring rapidly within the plume. These two processes lead to uncertainties in the estimation of the total bromine/sulfur ratio of the magmatic gas from $SO_2$ and BrO DOAS columns. This is why in situ measurements in volcanic fumaroles and plumes have generally been chosen to determine the total bromine/sulfur ratio used as input for the simulations of real case studies (Jourdain et al. 2016, Surl et al. 2021). Similarly, we choose this latter approach in our study to set the total bromine/sulphur ratio of the magmatic gas emissions.

The original version of the manuscript was not clear. This is not because $SO_2$ and BrO columns from remote sensing instruments were not available for this case study that we used in situ measurements but because in-situ measurements have an expected better accuracy for the estimation of the total bromine/sulphur ratio of the magmatic gas. We have simplified the text in the revised version by just explaining why we have used the in situ measurements for the magmatic gas composition.

Bromine emissions can be satisfactorily derived by in-situ direct sampling of both fumaroles (Gerlach, 2004) and plumes (Aiuppa et al., 2005), but both techniques are not viable measurement strategies in eruptive plumes due to the inherent risks for operators. We here therefore use the magmatic gas composition for the Etna's passive plume (Table 1) derived on 14 May 2008 by a combination of techniques (MultiGAS for $H_2O$, $CO_2$ and $SO_2$ and filter packs for halogens; see Aiuppa et al., 2005, 2007b, 2008 for analytical details). Note that previous modelling case studies of real volcanic emissions have also set the composition of the magmatic gas from in situ measurements (Jourdain et al. 2016, Surl et al. 2021). Here, the in situ data gathered on 14 May 2008 are used as an analogue for 10 May 2008 eruptive plume composition.

Dinger, F., Kleinbek, T., Dörner, S., Bobrowski, N., Platt, U., Wagner, T., Ibarra, M., and Espinoza, E.: $SO_2$ and BrO emissions of Masaya volcano from 2014 to 2020, Atmos. Chem. Phys., 21, 9367–9404, https://doi.org/10.5194/acp-21-9367-2021, 2021.

**- It is said (lines 730-731) that the bromine partition is realistic during the night. I did not see any nighttime measurements mentioned that can support this statement.**

This sentence is confusing. You are right to say that, at nighttime, there is no measurement available to compare with and thus to evaluate whether the model is realistic. What we meant is that the model provides the expected results considering that at nighttime $Br_2$ and BrCl formed from the heterogeneous reactions (reactions R5a and R5b) are no longer photolysed and thus bromine is mainly stored into the

Br$_2$ and BrCl reservoirs. We have changed the formulation of this sentence to take your remark into account.

During nighttime, the bromine explosion stops because there is no photolysis leading to bromine being mainly stored in the form of Br$_2$ and BrCl reservoirs as expected.

**If any additional experimental data are available, I suggest to show them in the Figures for comparison. If not, it may be better to make a general comparison between volcanic observations and the model instead of focusing on a case study for a specific Etna eruption.**

Since we do not have more experimental data available, we have changed the argument in the revised version for the choice of the case study. We now make more clearly the point that we aim to simulate a plausible case with realistic emissions and this is why we chose the case of 10 May 2008 for which the emissions of SO$_2$ and halogen compounds were available (at the beginning of section 2). But we no longer stress that the choice of the case study was because of the availability of the satellite observations. Still, we use the BrO and SO$_2$ columns and the BrO/SO$_2$ ratio from GOME-2 in addition to the literature to show that the model provides plausible values. As said before, we put less emphasis in the revised version on the model comparison with GOME-2 data.

**Specific comments:**

**- Page 4, line 112: It is unclear what is meant by the "explicit representation of Br2 species". There is only one Br2 species: molecular bromine. Did you mean "Bry" species instead of Br2?**

We mean Br$_2$. This adjective 'explicit' was used to emphasise that it was not taken into account in Grellier et al (2014) but it makes the sentence unclear. The adjective explicit was removed in the revised manuscript.

**- Why do you say on page 7 that there is "no mixing with background air" even though it is included when setting the X value to 0.1 or 0.3?**

There is mixing between the Plume box and the Model box when the Plume parameterization is used. But what we mean in lines 189-192 is that we assume that there is no exchange of air at the outside boundaries of the considered column (Model box). We have modified the text to make this clearer.

**- The caption of Table 3 does not explain the meaning of the X value, and when the table is mentioned in the text for the first time, X hasn't been mentioned yet.**

The figure caption has been changed to make clear that the X parameter is only used when the Plume parameterization is run and that the explanation of X is in section 4.2.

**- I first had the impression that the N.Ref simulation is identical to a simulation with X=0. Why, however, is the X value for N.Ref in Table 3 listed as "N/A" and not as "0"? Does this mean that a simulation with X=0 would be different from N.Ref? I checked "XFP" in the model code which seems to be the same as "1-X". As far as I can see, there is no difference between setting XFP=1 and PLUME2=.TRUE.**

Regarding the second part of your comment, you are right that in the code XFP=1-X and XFP=1 corresponds to X=0 and also to the PLUME2 case from Grellier et al. (2014). In this case, there is no mixing between the Plume box and the Model-P box during the duration of the eruption. This is at the time of the end of the eruption that the content of the Plume box is fully mixed with the Model-P box.

Regarding the first part of your comment, we think you meant X=1 and not X=0 since X=0 is the case that is described just above and that corresponds to the extreme case when there is no mixing at all

during the whole eruption and a full mixing at the end of the eruption. For the simulation with X=1, it is different from N.Ref. In the simulation N.Ref, the emissions are injected at each timestep in the Model box, meaning that they are directly diluted in the Model box and react with the molecules of all species present in the Model box. In the Plume simulation with X=1, the emissions are injected at each timestep in the Plume box. In practice, the molecules emitted are added to the molecules of all species present in the Plume box (which are 400 times less than in the Model box). Then the chemistry is applied to the Plume box and changes its composition. Finally, the content of the Plume box is fully mixed with the Model-P box at each timestep.
This information has been added in the revised manuscript in section 4.2.

**- The caption of Fig. 1 says that the Model-P Box is defined as the shaded blue square minus the big blue square. This would be a negative number. Is this correct?**

You are right, there was an error in the figure caption. The Model-P Box is the big blue square minus the shaded blue square. This has been corrected.

**- A vertical 1D model has no horizontal resolution. What do you mean on page 31 with "The 1D simulations were run with resolution of 0.5° longitude x 0.5° latitude"?**

This sentence was about the initialisation and was not clear. We have changed it in the revised manuscript. However, the horizontal size of the Model box needs to be set in our calculation since the Model box provides the volume in which the emissions are injected for the simulations without the plume parameterization. Moreover, the total burden of background oxidants, which play an important role in the bromine cycle, depends on the size of the Model box. In all simulations, assuming a larger model grid box (coarser horizontal resolution) provides a higher total burden of oxidants to react with the same volcanic emissions than in the case with a smaller grid box.

The 1D simulations were initialised from a MOCAGE 3D simulation with a resolution of 0.5° longitude x 0.5° latitude.

**- Page 3, line 89 and page 30, lines 717-719: The chemical lifetime of BrO is on the order of minutes. Therefore, it will not undergo long-range transport.**

You are right that BrO is a short-lived species. Within the volcanic plume, BrO is present during the whole daytime due to the bromine explosion. However, its primary loss mechanism, photolysis (BrO + hv -> Br + O3P), results in the formation of Br and the reformation of ozone, and, importantly, the loss of Br is entirely dominated by the Br + O3 -> reaction. Thus, during daytime, BrO continuously cycles back and forth between Br and BrO. At night-time, Br/BrO is stored in the $Br_2$ and BrCl reservoir species before being generated again during daytime. BrO, $Br_2$ and BrCl are insoluble species that do not undergo losses by wet deposition. This is why BrO can be found far from the volcano during daytime over several days after the emission until the plume dissipates. This was not clearly explained in the manuscript. In order not to go into too much details we have changed the sentence page 3 line 89.

'However, bromine emissions can be transported within the plume at regional scales (Jourdain et al. 2016, Narivelo et al. 2023).'
And we have removed the sentence P30 lines 717-719 and merged the sentences before and after.

Narivelo, H., Hamer, P. D., Marécal, V., Surl, L., Roberts, T., Pelletier, S., Josse, B., Guth, J., Bacles, M., Warnach, S., Wagner, T., Corradini, S., Salerno, G., and Guerrieri, L.: A regional modelling study of halogen chemistry within a volcanic plume of Mt Etna's Christmas 2018 eruption, EGUsphere [preprint], https://doi.org/10.5194/egusphere-2023-184, 2023.

**- The plots of BrO and BrO/SO2 in Fig. 2 are very similar. This means that SO2 is nearly constant, which makes sense for N.Ref. However, shouldn't SO2 decrease a lot via plume dilution during**

**the model runs N.Plume.0.1 and N.Plume.0.3? Can you add SO2 to the plots in Fig. 2? This would help to compare the dilution rates of SO2.**

All the figures show the results in the Model box. For the simulations with the Plume parameterization, the concentrations in the Model box come from adding the Model-P box and the Plume box concentrations (see L422-424 of the original manuscript). Since $SO_2$ mainly comes from the volcanic emissions and is only very slightly chemically depleted in both the Plume and the Model-P box, its total in the Model box is the same (left figure below). This is why the time variations of the $BrO/SO_2$ ratio are driven by BrO. This is now explained in the revised manuscript (section 5.1). We think this is not necessary to show the left figure below in the paper because it is not possible to distinguish the differences between the simulations.

The effect of the dilution can only be seen in the Plume box. We show in the right figure below the evolution of the concentration for a passive tracer having the same emission as $SO_2$ in the Plume box. This illustrates the dilution in the Plume box with time and relates to what was said in the original version of the paper (L364-365): X=0.3 and X=0.1 corresponds to a full dilution time of ~2.5 hours for X=0.3 and ~10 hours for X=0.1 after the end of the eruption, respectively. The Plume box is only used to calculate the chemical processing of the emissions within an air volume typical of the size of a volcanic plume. Since we are ultimately interested in analysing the effect of this processing on the final partitioning of the bromine species at the scale of the Model-box and to not confuse the reader, we prefer not to include the right figure below.

[Figure]

**- In Fig. 2 it can be seen that BrO starts to decrease even before the volcanic eruption emissions stop (i.e., inside the green zone). It would be interesting to explain this behaviour.**

This is because the amount of sunlight starts to decrease significantly in the two timesteps before the end of the eruption, which occur shortly before twilight. This strongly reduces the efficiency of the bromine explosion because of the weakening of the photolysis of $Br_2$ and BrCl. This is also why $Br_2$, and to a lesser extent BrCl, increase during those timesteps (see Fig. 3a). It was explained in lines 433-434. The text has been revised with a more detailed explanation.

After 17:45 UTC and before the full night, the daylight starts to decrease significantly and this strongly reduces the efficiency of the bromine explosion even if there are still bromine emissions. This is linked to a weakening of the photolysis of $Br_2$ and BrCl. This is also why $Br_2$, and to a lesser extent BrCl, increase during those timesteps (see Fig. 3a).

Citation: https://doi.org/10.5194/gmd-2022-180-RC2